# Variational Diffusion Models

**Diederik P. Kingma**\*, **Tim Salimans**\*, **Ben Poole, Jonathan Ho**
Google Research

## Abstract

Diffusion-based generative models have demonstrated a capacity for perceptually impressive synthesis, but can they also be great likelihood-based models? We answer this in the affirmative, and introduce a family of diffusion-based generative models that obtain state-of-the-art likelihoods on standard image density estimation benchmarks. Unlike other diffusion-based models, our method allows for efficient optimization of the noise schedule jointly with the rest of the model. We show that the variational lower bound (VLB) simplifies to a remarkably short expression in terms of the signal-to-noise ratio of the diffused data, thereby improving our theoretical understanding of this model class. Using this insight, we prove an equivalence between several models proposed in the literature. In addition, we show that the continuous-time VLB is invariant to the noise schedule, except for the signal-to-noise ratio at its endpoints. This enables us to learn a noise schedule that minimizes the variance of the resulting VLB estimator, leading to faster optimization. Combining these advances with architectural improvements, we obtain state-of-the-art likelihoods on image density estimation benchmarks, outperforming autoregressive models that have dominated these benchmarks for many years, with often significantly faster optimization. In addition, we show how to use the model as part of a bits-back compression scheme, and demonstrate lossless compression rates close to the theoretical optimum.

## 1 Introduction

Likelihood-based generative modeling is a central task in machine learning that is the basis for a wide range of applications ranging from speech synthesis [Oord et al., 2016], to translation [Sutskever et al., 2014], to compression [MacKay, 2003], to many others. Autoregressive models have long been the dominant model class on this task due to their tractable likelihood and expressivity, as shown in Figure 1. Diffusion models have recently shown impressive results in image [Ho et al., 2020, Song et al., 2021b, Nichol and Dhariwal, 2021] and audio generation [Kong et al., 2020, Chen et al., 2020] in terms of perceptual quality, but have yet to match autoregressive models on density estimation benchmarks. In this paper we make several technical contributions that allow diffusion models to challenge the dominance of autoregressive models in this domain. Our main contributions are as follows:

- We introduce a flexible family of diffusion-based generative models that achieve new state-of-the-art log-likelihoods on standard image density estimation benchmarks (CIFAR-10 and ImageNet). This is enabled by incorporating Fourier features into the diffusion model and using a learnable specification of the diffusion process, among other modeling innovations.

- We improve our theoretical understanding of density modeling using diffusion models by analyzing their variational lower bound (VLB), deriving a remarkably simple expression in terms of the signal-to-noise ratio of the diffusion process. This result delivers new insight

---

\* Equal contribution.

35th Conference on Neural Information Processing Systems (NeurIPS 2021).

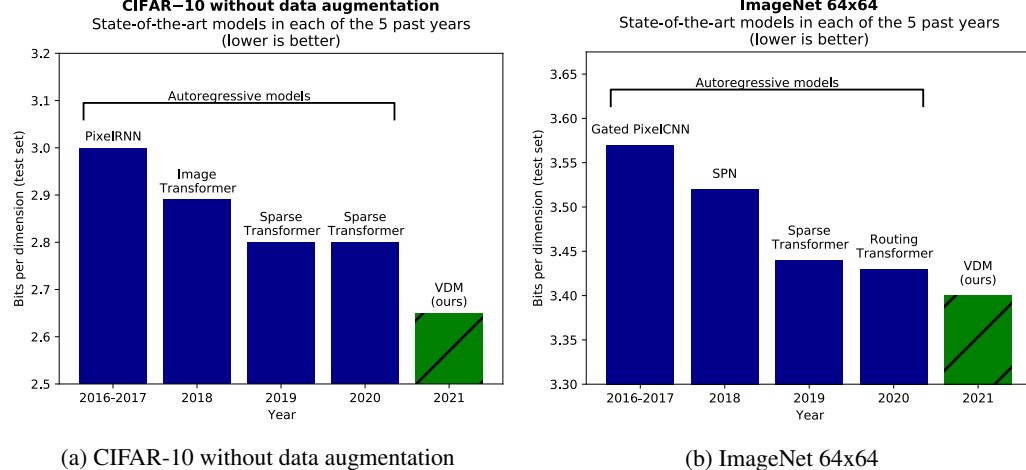

(a) CIFAR-10 without data augmentation          (b) ImageNet 64x64

Figure 1: Autoregressive generative models were long dominant in standard image density estimation benchmarks. In contrast, we propose a family of diffusion-based generative models, *Variational Diffusion Models* (VDMs), that outperforms contemporary autoregressive models in these benchmarks. See Table 1 for more results and comparisons.

into the model class: for the continuous-time (infinite-depth) setting we prove a novel invariance of the generative model and its VLB to the specification of the diffusion process, and we show that various diffusion models from the literature are equivalent up to a trivial time-dependent rescaling of the data.

## 2   Related work

Our work builds on diffusion probabilistic models (DPMs) [Sohl-Dickstein et al., 2015], or *diffusion models* in short. DPMs can be viewed as a type of variational autoencoder (VAE) [Kingma and Welling, 2013, Rezende et al., 2014], whose structure and loss function allows for efficient training of arbitrarily deep models. Interest in diffusion models has recently reignited due to their impressive image generation results [Ho et al., 2020, Song and Ermon, 2020].

Ho et al. [2020] introduced a number of model innovations to the original DPM, with impressive results on image generation quality benchmarks. They showed that the VLB objective, for a diffusion model with discrete time and diffusion variances shared across input dimensions, is equivalent to multi-scale denoising score matching, up to particular weightings per noise scale. Further improvements were proposed by Nichol and Dhariwal [2021], resulting in better log-likelihood scores. Gao et al. [2020] show how diffusion can also be used to efficiently optimize energy-based models (EBMs) towards a close approximation of the log-likelihood objective, resulting in high-fidelity samples even after long MCMC chains.

Song and Ermon [2019] first proposed learning generative models through a multi-scale denoising score matching objective, with improved methods in Song and Ermon [2020]. This was later extended to continuous-time diffusion with novel sampling algorithms based on reversing the diffusion process [Song et al., 2021b].

Concurrent to our work, Song et al. [2021a], Huang et al. [2021], and Vahdat et al. [2021] also derived variational lower bounds to the data likelihood under a continuous-time diffusion model. Where we consider the infinitely deep limit of a standard VAE, Song et al. [2021a] and Vahdat et al. [2021] present different derivations based on stochastic differential equations. Huang et al. [2021] considers both perspectives and discusses the similarities between the two approaches. An advantage of our analysis compared to these other works is that we present an intuitive expression of the VLB in terms of the signal-to-noise ratio of the diffused data, leading to much simplified expressions of the discrete-time and continuous-time loss, allowing for simple and numerically stable implementation. This also leads to new results on the invariance of the generative model and its VLB

to the specification of the diffusion process. We empirically compare to these works, as well as others, in Table 1.

Previous approaches to diffusion probabilistic models fixed the diffusion process; in contrast optimize the diffusion process parameters jointly with the rest of the model. This turns the model into a type of VAE [Kingma and Welling, 2013, Rezende et al., 2014]. This is enabled by directly parameterizing the mean and variance of the marginal $q(\mathbf{z}_t|\mathbf{z}_0)$, where previous approaches instead parameterized the individual diffusion steps $q(\mathbf{z}_{t+\epsilon}|\mathbf{z}_t)$. In addition, our denoising models include several architecture changes, the most important of which is the use of Fourier features, which enable us to reach much better likelihoods than previous diffusion probabilistic models.

## 3 Model

We will focus on the most basic case of generative modeling, where we have a dataset of observations of $\mathbf{x}$, and the task is to estimate the marginal distribution $p(\mathbf{x})$. As with most generative models, the described methods can be extended to the case of multiple observed variables, and/or the task of estimating conditional densities $p(\mathbf{x}|\mathbf{y})$. The proposed latent-variable model consists of a diffusion process (Section 3.1) that we invert to obtain a hierarchical generative model (Section 3.3). As we will show, the model choices below result in a surprisingly simple variational lower bound (VLB) of the marginal likelihood, which we use for optimization of the parameters.

### 3.1 Forward time diffusion process

Our starting point is a Gaussian diffusion process that begins with the data $\mathbf{x}$, and defines a sequence of increasingly noisy versions of $\mathbf{x}$ which we call the *latent variables* $\mathbf{z}_t$, where $t$ runs from $t = 0$ (least noisy) to $t = 1$ (most noisy). The distribution of latent variable $\mathbf{z}_t$ conditioned on $\mathbf{x}$, for any $t \in [0, 1]$ is given by:

$$q(\mathbf{z}_t|\mathbf{x}) = \mathcal{N}\left(\alpha_t \mathbf{x}, \sigma_t^2 \mathbf{I}\right), \tag{1}$$

where $\alpha_t$ and $\sigma_t^2$ are strictly positive scalar-valued functions of $t$. Furthermore, let us define the *signal-to-noise ratio* (SNR):

$$\text{SNR}(t) = \alpha_t^2/\sigma_t^2. \tag{2}$$

We assume that the $\text{SNR}(t)$ is strictly monotonically decreasing in time, i.e. that $\text{SNR}(t) < \text{SNR}(s)$ for any $t > s$. This formalizes the notion that the $\mathbf{z}_t$ is increasingly noisy as we go forward in time. We also assume that both $\alpha_t$ and $\sigma_t^2$ are smooth, such that their derivatives with respect to time $t$ are finite. This diffusion process specification includes the *variance-preserving* diffusion process as used by [Sohl-Dickstein et al., 2015, Ho et al., 2020] as a special case, where $\alpha_t = \sqrt{1 - \sigma_t^2}$. Another special case is the *variance-exploding* diffusion process as used by [Song and Ermon, 2019, Song et al., 2021b], where $\alpha_t^2 = 1$. In experiments, we use the variance-preserving version.

The distributions $q(\mathbf{z}_t|\mathbf{z}_s)$ for any $t > s$ are also Gaussian, and given in Appendix A. The joint distribution of latent variables $(\mathbf{z}_s, \mathbf{z}_t, \mathbf{z}_u)$ at any subsequent timesteps $0 \leq s < t < u \leq 1$ is Markov: $q(\mathbf{z}_u|\mathbf{z}_t, \mathbf{z}_s) = q(\mathbf{z}_u|\mathbf{z}_t)$. Given the distributions above, it is relatively straightforward to verify through Bayes rule that $q(\mathbf{z}_s|\mathbf{z}_t, \mathbf{x})$, for any $0 \leq s < t \leq 1$, is also Gaussian. This distribution is also given in Appendix A.

### 3.2 Noise schedule

In previous work, the noise schedule has a fixed form (see Appendix H, Fig. 4a). In contrast, we learn this schedule through the parameterization

$$\sigma_t^2 = \text{sigmoid}(\gamma_{\boldsymbol{\eta}}(t)) \tag{3}$$

where $\gamma_{\boldsymbol{\eta}}(t)$ is a monotonic neural network with parameters $\boldsymbol{\eta}$, as detailed in Appendix H.

Motivated by the equivalence discussed in Section 5.1, we use $\alpha_t = \sqrt{1 - \sigma_t^2}$ in our experiments for both the discrete-time and continuous-time models, i.e. variance-preserving diffusion processes. It is straightforward to verify that $\alpha_t^2$ and $\text{SNR}(t)$, as a function of $\gamma_{\boldsymbol{\eta}}(t)$, then simplify to:

$$\alpha_t^2 = \text{sigmoid}(-\gamma_{\boldsymbol{\eta}}(t)) \tag{4}$$
$$\text{SNR}(t) = \exp(-\gamma_{\boldsymbol{\eta}}(t)) \tag{5}$$

## 3.3 Reverse time generative model

We define our generative model by inverting the diffusion process of Section 3.1, yielding a hierarchical generative model that samples a sequence of latents $\mathbf{z}_t$, with time running backward from $t = 1$ to $t = 0$. We consider both the case where this sequence consists of a finite number of steps $T$, as well as a continuous time model corresponding to $T \to \infty$. We start by presenting the discrete-time case.

Given finite $T$, we discretize time uniformly into $T$ timesteps (segments) of width $\tau = 1/T$. Defining $s(i) = (i-1)/T$ and $t(i) = i/T$, our hierarchical generative model for data $\mathbf{x}$ is then given by:

$$p(\mathbf{x}) = \int_{\mathbf{z}} p(\mathbf{z}_1) p(\mathbf{x}|\mathbf{z}_0) \prod_{i=1}^{T} p(\mathbf{z}_{s(i)}|\mathbf{z}_{t(i)}). \tag{6}$$

With the variance preserving diffusion specification and sufficiently small $\text{SNR}(1)$, we have that $q(\mathbf{z}_1|\mathbf{x}) \approx \mathcal{N}(\mathbf{z}_1; 0, \mathbf{I})$. We therefore model the marginal distribution of $\mathbf{z}_1$ as a spherical Gaussian:

$$p(\mathbf{z}_1) = \mathcal{N}(\mathbf{z}_1; 0, \mathbf{I}). \tag{7}$$

We wish to choose a model $p(\mathbf{x}|\mathbf{z}_0)$ that is close to the unknown $q(\mathbf{x}|\mathbf{z}_0)$. Let $x_i$ and $z_{0,i}$ be the $i$-th elements of $\mathbf{x}, \mathbf{z}_0$, respectively. We then use a factorized distribution of the form:

$$p(\mathbf{x}|\mathbf{z}_0) = \prod_i p(x_i|z_{0,i}), \tag{8}$$

where we choose $p(x_i|z_{0,i}) \propto q(z_{0,i}|x_i)$, which is normalized by summing over all possible discrete values of $x_i$ (256 in the case of 8-bit image data). With sufficiently large $\text{SNR}(0)$, this becomes a very close approximation to the true $q(\mathbf{x}|\mathbf{z}_0)$, as the influence of the unknown data distribution $q(\mathbf{x})$ is overwhelmed by the likelihood $q(\mathbf{z}_0|\mathbf{x})$. Finally, we choose the conditional model distributions as

$$p(\mathbf{z}_s|\mathbf{z}_t) = q(\mathbf{z}_s|\mathbf{z}_t, \mathbf{x} = \hat{\mathbf{x}}_{\boldsymbol{\theta}}(\mathbf{z}_t; t)), \tag{9}$$

i.e. the same as $q(\mathbf{z}_s|\mathbf{z}_t, \mathbf{x})$, but with the original data $\mathbf{x}$ replaced by the output of a *denoising model* $\hat{\mathbf{x}}_{\boldsymbol{\theta}}(\mathbf{z}_t; t)$ that predicts $\mathbf{x}$ from its noisy version $\mathbf{z}_t$. Note that in practice we parameterize the denoising model as a function of a *noise prediction model* (Section 3.4), bridging the gap with previous work on diffusion models [Ho et al., 2020]. The means and variances of $p(\mathbf{z}_s|\mathbf{z}_t)$ simplify to a remarkable degree; see Appendix A.

## 3.4 Noise prediction model and Fourier features

We parameterize the denoising model in terms of a *noise prediction model* $\hat{\boldsymbol{\epsilon}}_{\boldsymbol{\theta}}(\mathbf{z}_t; t)$:

$$\hat{\mathbf{x}}_{\boldsymbol{\theta}}(\mathbf{z}_t; t) = (\mathbf{z}_t - \sigma_t \hat{\boldsymbol{\epsilon}}_{\boldsymbol{\theta}}(\mathbf{z}_t; t))/\alpha_t, \tag{10}$$

where $\hat{\boldsymbol{\epsilon}}_{\boldsymbol{\theta}}(\mathbf{z}_t; t)$ is parameterized as a neural network. The noise prediction models we use in experiments closely follow Ho et al. [2020], except that they process the data solely at the original resolution. The exact parameterization of the noise prediction model and noise schedule is discussed in Appendix B.

Prior work on diffusion models has mainly focused on the perceptual quality of generated samples, which emphasizes coarse scale patterns and global consistency of generated images. Here, we optimize for likelihood, which is sensitive to fine scale details and exact values of individual pixels. To capture the fine scale details of the data, we propose adding a set of *Fourier features* to the input of our noise prediction model. Let $\mathbf{x}$ be the original data, scaled to the range $[-1, 1]$, and let $\mathbf{z}$ be the resulting latent variable, with similar magnitudes. We then append channels $\sin(2^n \pi \mathbf{z})$ and $\cos(2^n \pi \mathbf{z})$, where $n$ runs over a range of integers $\{n_{min}, ..., n_{max}\}$. These features are high frequency periodic functions that amplify small changes in the input data $\mathbf{z}_t$; see Appendix C for further details. Including these features in the input of our denoising model leads to large improvements in likelihood as demonstrated in Section 6 and Figure 5, especially when combined with a learnable SNR function. We did not observe such improvements when incorporating Fourier features into autoregressive models.

## 3.5 Variational lower bound

We optimize the parameters towards the variational lower bound (VLB) of the marginal likelihood, which is given by

$$-\log p(\mathbf{x}) \leq -\text{VLB}(\mathbf{x}) = \underbrace{D_{KL}(q(\mathbf{z}_1|\mathbf{x})\|p(\mathbf{z}_1))}_{\text{Prior loss}} + \underbrace{\mathbb{E}_{q(\mathbf{z}_0|\mathbf{x})}\left[-\log p(\mathbf{x}|\mathbf{z}_0)\right]}_{\text{Reconstruction loss}} + \underbrace{\mathcal{L}_T(\mathbf{x})}_{\text{Diffusion loss}}. \qquad (11)$$

The prior loss and reconstruction loss can be (stochastically and differentiably) estimated using standard techniques; see [Kingma and Welling, 2013]. The diffusion loss, $\mathcal{L}_T(\mathbf{x})$, is more complicated, and depends on the hyperparameter $T$ that determines the depth of the generative model.

## 4 Discrete-time model

In the case of finite $T$, using $s(i) = (i-1)/T$, $t(i) = i/T$, the diffusion loss is:

$$\mathcal{L}_T(\mathbf{x}) = \sum_{i=1}^{T} \mathbb{E}_{q(\mathbf{z}_{t(i)}|\mathbf{x})} D_{KL}[q(\mathbf{z}_{s(i)}|\mathbf{z}_{t(i)}, \mathbf{x})\|p(\mathbf{z}_{s(i)}|\mathbf{z}_{t(i)})]. \qquad (12)$$

In appendix E we show that this expression simplifies considerably, yielding:

$$\mathcal{L}_T(\mathbf{x}) = \frac{T}{2}\mathbb{E}_{\boldsymbol{\epsilon}\sim\mathcal{N}(0,\mathbf{I}),i\sim U\{1,T\}}\left[(\text{SNR}(s) - \text{SNR}(t))\|\mathbf{x} - \hat{\mathbf{x}}_{\boldsymbol{\theta}}(\mathbf{z}_t; t)\|_2^2\right], \qquad (13)$$

where $U\{1, T\}$ is the uniform distribution on the integers $\{1, \ldots, T\}$, and $\mathbf{z}_t = \alpha_t\mathbf{x} + \sigma_t\boldsymbol{\epsilon}$. This is the general discrete-time loss for any choice of forward diffusion parameters $(\sigma_t, \alpha_t)$. When plugging in the specifications of $\sigma_t$, $\alpha_t$ and $\hat{\mathbf{x}}_{\boldsymbol{\theta}}(\mathbf{z}_t; t)$ that we use in experiments, given in Sections 3.2 and 3.4, the loss simplifies to:

$$\mathcal{L}_T(\mathbf{x}) = \frac{T}{2}\mathbb{E}_{\boldsymbol{\epsilon}\sim\mathcal{N}(0,\mathbf{I}),i\sim U\{1,T\}}\left[(\exp(\gamma_{\boldsymbol{\eta}}(t) - \gamma_{\boldsymbol{\eta}}(s)) - 1)\|\boldsymbol{\epsilon} - \hat{\boldsymbol{\epsilon}}_{\boldsymbol{\theta}}(\mathbf{z}_t; t)\|_2^2\right] \qquad (14)$$

where $\mathbf{z}_t = \text{sigmoid}(-\gamma_{\boldsymbol{\eta}}(t))\mathbf{x} + \text{sigmoid}(\gamma_{\boldsymbol{\eta}}(t))\boldsymbol{\epsilon}$. In the discrete-time case, we simply jointly optimize $\boldsymbol{\eta}$ and $\boldsymbol{\theta}$ by maximizing the VLB through a Monte Carlo estimator of Equation 14.

Note that $\exp(.) - 1$ has a numerically stable primitive $expm1(.)$ in common numerical computing packages; see figure 6. Equation 14 allows for numerically stable implementation in 32-bit or lower-precision floating point, in contrast with previous implementations of discrete-time diffusion models (e.g. [Ho et al., 2020]), which had to resort to 64-bit floating point.

### 4.1 More steps leads to a lower loss

A natural question to ask is what the number of timesteps $T$ should be, and whether more timesteps is always better in terms of the VLB. In Appendix F we analyze the difference between the diffusion loss with $T$ timesteps, $\mathcal{L}_T(\mathbf{x})$, and the diffusion loss with double the timesteps, $\mathcal{L}_{2T}(\mathbf{x})$, while keeping the SNR function fixed. We then find that if our trained denoising model $\hat{\mathbf{x}}_{\boldsymbol{\theta}}$ is sufficiently good, we have that $\mathcal{L}_{2T}(\mathbf{x}) < \mathcal{L}_T(\mathbf{x})$, i.e. that our VLB will be better for a larger number of timesteps. Intuitively, the discrete time diffusion loss is an upper Riemann sum approximation of an integral of a strictly decreasing function, meaning that a finer approximation yields a lower diffusion loss. This result is illustrated in Figure 2.

## 5 Continuous-time model: $T \to \infty$

Since taking more time steps leads to a better VLB, we now take $T \to \infty$, effectively treating time $t$ as continuous rather than discrete. The model for $p(\mathbf{z}_t)$ can in this case be described as a continuous time diffusion process [Song et al., 2021b] governed by a stochastic differential equation; see Appendix D. In Appendix E we show that in this limit the diffusion loss $\mathcal{L}_T(\mathbf{x})$ simplifies further. Letting $\text{SNR}'(t) = d\text{SNR}(t)/dt$, we have, with $\mathbf{z}_t = \alpha_t\mathbf{x} + \sigma_t\boldsymbol{\epsilon}$:

$$\mathcal{L}_\infty(\mathbf{x}) = -\frac{1}{2}\mathbb{E}_{\boldsymbol{\epsilon}\sim\mathcal{N}(0,\mathbf{I})}\int_0^1 \text{SNR}'(t)\|\mathbf{x} - \hat{\mathbf{x}}_{\boldsymbol{\theta}}(\mathbf{z}_t; t)\|_2^2\, dt, \qquad (15)$$

$$= -\frac{1}{2}\mathbb{E}_{\boldsymbol{\epsilon}\sim\mathcal{N}(0,\mathbf{I}),t\sim\mathcal{U}(0,1)}\left[\text{SNR}'(t)\|\mathbf{x} - \hat{\mathbf{x}}_{\boldsymbol{\theta}}(\mathbf{z}_t; t)\|_2^2\right]. \qquad (16)$$

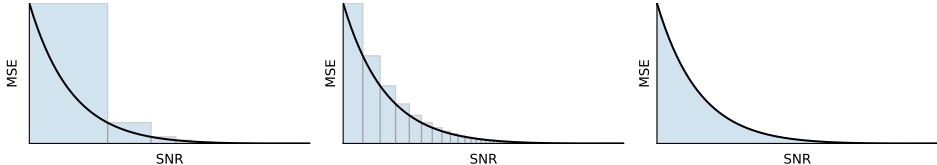

Figure 2: Illustration of the diffusion loss with few segments $T$ (left), more segments $T$ (middle), and infinite segments $T$ (continuous time, right). The continuous-time loss (Equation 18) is an integral of mean squared error (MSE) over SNR, here visualized as a black curve. The black curve is strictly decreasing when the model is sufficiently well trained, so the discrete-time loss (Equation 13) is an upper bound (an upper Riemann sum approximation) of this integral that becomes better when segments are added.

This is the general continuous-time loss for any choice of forward diffusion parameters $(\sigma_t, \alpha_t)$. When plugging in the specifications of $\sigma_t$, $\alpha_t$ and $\hat{\mathbf{x}}_{\boldsymbol{\theta}}(\mathbf{z}_t; t)$ that we use in experiments, given in Sections 3.2 and 3.4, the loss simplifies to:

$$\mathcal{L}_\infty(\mathbf{x}) = \frac{1}{2}\mathbb{E}_{\boldsymbol{\epsilon}\sim\mathcal{N}(0,\mathbf{I}),t\sim\mathcal{U}(0,1)}\left[\gamma'_{\boldsymbol{\eta}}(t)\left\|\boldsymbol{\epsilon}-\hat{\boldsymbol{\epsilon}}_{\boldsymbol{\theta}}(\mathbf{z}_t;t)\right\|_2^2\right], \tag{17}$$

where $\gamma'_{\boldsymbol{\eta}}(t) = d\gamma_{\boldsymbol{\eta}}(t)/dt$. We use the Monte Carlo estimator of this loss for evaluation and optimization.

## 5.1 Equivalence of diffusion models in continuous time

The signal-to-noise function $\text{SNR}(t)$ is invertible due to the monotonicity assumption in Section 3.1. Due to this invertibility, we can perform a change of variables, and make everything a function of $v \equiv \text{SNR}(t)$ instead of $t$, such that $t = \text{SNR}^{-1}(v)$. Let $\alpha_v$ and $\sigma_v$ be the functions $\alpha_t$ and $\sigma_t$ evaluated at $t = \text{SNR}^{-1}(v)$, and correspondingly let $\mathbf{z}_v = \alpha_v\mathbf{x} + \sigma_v\boldsymbol{\epsilon}$. Similarly, we rewrite our noise prediction model as $\tilde{\mathbf{x}}_{\boldsymbol{\theta}}(\mathbf{z}, v) \equiv \hat{\mathbf{x}}_{\boldsymbol{\theta}}(\mathbf{z}, \text{SNR}^{-1}(v))$. With this change of variables, our continuous-time loss in Equation 15 can equivalently be written as:

$$\mathcal{L}_\infty(\mathbf{x}) = \frac{1}{2}\mathbb{E}_{\boldsymbol{\epsilon}\sim\mathcal{N}(0,\mathbf{I})}\int_{\text{SNR}_{\min}}^{\text{SNR}_{\max}}\left\|\mathbf{x}-\tilde{\mathbf{x}}_{\boldsymbol{\theta}}(\mathbf{z}_v, v)\right\|_2^2 dv, \tag{18}$$

where instead of integrating w.r.t. time $t$ we now integrate w.r.t. the signal-to-noise ratio $v$, and where $\text{SNR}_{\min} = \text{SNR}(1)$ and $\text{SNR}_{\max} = \text{SNR}(0)$.

What this equation shows us is that the only effect the functions $\alpha(t)$ and $\sigma(t)$ have on the diffusion loss is through the values $\text{SNR}(t) = \alpha_t^2/\sigma_t^2$ at endpoints $t = 0$ and $t = 1$. Given these values $\text{SNR}_{\max}$ and $\text{SNR}_{\min}$, the diffusion loss is invariant to the shape of function $\text{SNR}(t)$ between $t = 0$ and $t = 1$. The VLB is thus only impacted by the function $\text{SNR}(t)$ through its endpoints $\text{SNR}_{\min}$ and $\text{SNR}_{\max}$.

Furthermore, we find that the distribution $p(\mathbf{x})$ defined by our generative model is also invariant to the specification of the diffusion process. Specifically, let $p^A(\mathbf{x})$ denote the distribution defined by the combination of a diffusion specification and denoising function $\{\alpha_v^A, \sigma_v^A, \tilde{\mathbf{x}}_{\boldsymbol{\theta}}^A\}$, and similarly let $p^B(\mathbf{x})$ be the distribution defined through a different specification $\{\alpha_v^B, \sigma_v^B, \tilde{\mathbf{x}}_{\boldsymbol{\theta}}^B\}$, where both specifications have equal $\text{SNR}_{\min}, \text{SNR}_{\max}$; as shown in Appendix G, we then have that $p^A(\mathbf{x}) = p^B(\mathbf{x})$ if $\tilde{\mathbf{x}}_{\boldsymbol{\theta}}^B(\mathbf{z}, v) \equiv \tilde{\mathbf{x}}_{\boldsymbol{\theta}}^A((\alpha_v^A/\alpha_v^B)\mathbf{z}, v)$. The distribution on all latents $\mathbf{z}_v$ is then also the same under both specifications, up to a trivial rescaling. This means that any two diffusion models satisfying the mild constraints set in 3.1 (which includes e.g. the *variance exploding* and *variance preserving* specifications considered by Song et al. [2021b]), can thus be seen as equivalent in continuous time.

## 5.2 Weighted diffusion loss

This equivalence between diffusion specifications continues to hold even if, instead of the VLB, these models optimize a *weighted* diffusion loss of the form:

$$\mathcal{L}_\infty(\mathbf{x}, w) = \frac{1}{2}\mathbb{E}_{\boldsymbol{\epsilon}\sim\mathcal{N}(0,\mathbf{I})}\int_{\text{SNR}_{\min}}^{\text{SNR}_{\max}} w(v)\left\|\mathbf{x}-\tilde{\mathbf{x}}_{\boldsymbol{\theta}}(\mathbf{z}_v, v)\right\|_2^2 dv, \tag{19}$$

| Model (Bits per dim on test set) | Type | CIFAR10 no data aug. | CIFAR10 data aug. | ImageNet 32x32 | ImageNet 64x64 |
|---|---|---|---|---|---|
| *Previous work* | | | | | |
| ResNet VAE with IAF [Kingma et al., 2016] | VAE | 3.11 | | | |
| Very Deep VAE [Child, 2020] | VAE | 2.87 | | 3.80 | 3.52 |
| NVAE [Vahdat and Kautz, 2020] | VAE | 2.91 | | 3.92 | |
| Glow [Kingma and Dhariwal, 2018] | Flow | | $3.35^{(B)}$ | 4.09 | 3.81 |
| Flow++ [Ho et al., 2019a] | Flow | 3.08 | | 3.86 | 3.69 |
| PixelCNN [Van Oord et al., 2016] | AR | 3.03 | | 3.83 | 3.57 |
| PixelCNN++ [Salimans et al., 2017] | AR | 2.92 | | | |
| Image Transformer [Parmar et al., 2018] | AR | 2.90 | | 3.77 | |
| SPN [Menick and Kalchbrenner, 2018] | AR | | | | 3.52 |
| Sparse Transformer [Child et al., 2019] | AR | 2.80 | | | 3.44 |
| Routing Transformer [Roy et al., 2021] | AR | | | | 3.43 |
| Sparse Transformer + DistAug [Jun et al., 2020] | AR | | $2.53^{(A)}$ | | |
| DDPM [Ho et al., 2020] | Diff | | $3.69^{(C)}$ | | |
| EBM-DRL [Gao et al., 2020] | Diff | | $3.18^{(C)}$ | | |
| Score SDE [Song et al., 2021b] | Diff | 2.99 | | | |
| Improved DDPM [Nichol and Dhariwal, 2021] | Diff | 2.94 | | | 3.54 |
| *Concurrent work* | | | | | |
| CR-NVAE [Sinha and Dieng, 2021] | VAE | | $2.51^{(A)}$ | | |
| LSGM [Vahdat et al., 2021] | Diff | 2.87 | | | |
| ScoreFlow [Song et al., 2021a] (variational bound) | Diff | | $2.90^{(C)}$ | 3.86 | |
| ScoreFlow [Song et al., 2021a] (cont. norm. flow) | Diff | 2.83 | $2.80^{(C)}$ | 3.76 | |
| *Our work* | | | | | |
| **VDM (variational bound)** | Diff | **2.65** | $\mathbf{2.49}^{(A)}$ | **3.72** | **3.40** |

Table 1: Summary of our findings for density modeling tasks, in terms of bits per dimension (BPD) on the test set. Model types are autoregressive (AR), normalizing flows (Flow), variational autoencoders (VAE), or diffusion models (Diff). Our results were obtained using the continuous-time formulation of our model. CIFAR-10 data augmentations are: (A) extensive, (B) small translations, or (C) horizontal flips. The numbers for VDM are variational bounds, and can likely be improved by estimating the marginal likelihood through importance sampling, or through evaluation of the corresponding continuous normalizing flow as done by Song et al. [2021a].

which e.g. captures all the different objectives discussed by Song et al. [2021b], see Appendix K. Here, $w(v)$ is a weighting function that generally puts increased emphasis on the noisier data compared to the VLB, and which thereby can sometimes improve perceptual generation quality as measured by certain metrics like FID and Inception Score. For the models presented in this paper, we further use $w(v) = 1$ as corresponding to the (unweighted) VLB.

## 5.3 Variance minimization

Lowering the variance of the Monte Carlo estimator of the continuous-time loss generally improves the efficiency of optimization. We found that using a low-discrepancy sampler for $t$, as explained in Appendix I.1, leads to a significant reduction in variance. In addition, due to the invariance shown in Section 5.1 for the continous-time case, we can optimize the schedule *between* its endpoints w.r.t. to minimize the variance of our estimator of loss, as detailed in Appendix I. The endpoints of the noise schedule are simply optimized w.r.t. the VLB.

## 6 Experiments

We demonstrate our proposed class of diffusion models, which we call *Variational Diffusion Models* (VDMs), on the CIFAR-10 [Krizhevsky et al., 2009] dataset, and the downsampled ImageNet [Van Oord et al., 2016, Deng et al., 2009] dataset, where we focus on maximizing likelihood. For our result with data augmentation we used random flips, 90-degree rotations, and color channel swapping. More details of our model specifications are in Appendix B.

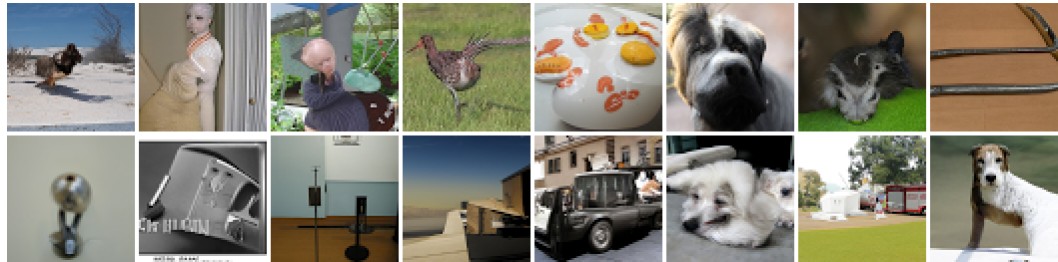

Figure 3: Non cherry-picked unconditional samples from our Imagenet 64x64 model, trained in continuous time and generated using $T = 1000$. The model's hyper-parameters and parameters are optimized w.r.t. the likelihood bound, so the model is not optimized for synthesis quality.

## 6.1 Likelihood and samples

Table 1 shows our results on modeling the CIFAR-10 dataset, and the downsampled ImageNet dataset. We establish a new state-of-the-art in terms of test set likelihood on all considered benchmarks, by a significant margin. Our model for CIFAR-10 without data augmentation surpasses the previous best result of 2.80 about 10x faster than it takes the Sparse Transformer to reach this, in wall clock time on equivalent hardware. Our CIFAR-10 model, whose hyper-parameters were tuned for likelihood, results in a FID (perceptual quality) score of 7.41. This would have been state-of-the-art until recently, but is worse than recent diffusion models that specifically target FID scores [Nichol and Dhariwal, 2021, Song et al., 2021b, Ho et al., 2020]. By instead using a weighted diffusion loss, with the weighting function $w(\text{SNR})$ used by Ho et al. [2020] and described in Appendix K, our FID score improves to 4.0. We did not pursue further tuning of the model to improve FID instead of likelihood. A random sample of generated images from our model is provided in Figure 3. We provide additional samples from this model, as well as our other models for the other datasets, in Appendix M.

## 6.2 Ablations

Next, we investigate the relative importance of our contributions. In Table 2 we compare our discrete-time and continuous-time specifications of the diffusion model: When evaluating our model with a small number of steps, our discretely trained models perform better by learning the diffusion schedule to optimize the VLB. However, as argued theoretically in Section 4.1, we find experimentally that more steps $T$ indeed gives better likelihood. When $T$ grows large, our continuously trained model performs best, helped by training its diffusion schedule to minimize variance instead.

Minimizing the variance also helps the continuous time model to train faster, as shown in Figure 5. This effect is further examined in Table 4b, where we find dramatic variance reductions compared to our baselines in continuous time. Figure 4a shows how this effect is achieved: Compared to the other schedules, our learned schedule spends much more time in the high $\text{SNR}(t)$ / low $\sigma_t^2$ range.

In Figure 5 we further show training curves for our model including and excluding the Fourier features proposed in Appendix C: with Fourier features enabled our model achieves much better likelihood. For comparison we also implemented Fourier features in a PixelCNN++ model [Salimans et al., 2017], where we do not see a benefit. In addition, we find that learning the SNR is necessary to get the most out of including Fourier features: if we fix the SNR schedule to that used by Ho et al. [2020], the maximum log-SNR is fixed to approximately 8 (see figure 7), and test set negative likelihood stays above 4 bits per dim. When learning the SNR endpoints, our maximum log-SNR ends up at 13.3, which, combined with the inclusion of Fourier features, leads to the SOTA test set likelihoods reported in Table 1.

## 6.3 Lossless compression

For a fixed number of evaluation timesteps $T_{eval}$, our diffusion model in discrete time is a hierarchical latent variable model that can be turned into a lossless compression algorithm using bits-back coding [Hinton and Van Camp, 1993]. As a proof of concept of practical lossless compression, Table 2 reports net codelengths on the CIFAR10 test set for various settings of $T_{eval}$ using BB-

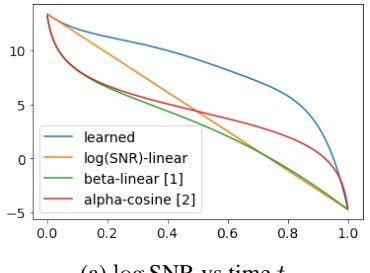

(a) $\log$ SNR vs time $t$

| SNR$(t)$ schedule | Var(BPD) |
|---|---|
| **Learned (ours)** | **0.53** |
| $\log$ SNR-linear | 6.35 |
| $\beta$-Linear [1] | 31.6 |
| $\alpha$-Cosine [2] | 31.1 |

(b) Variance of VLB estimate

Figure 4: Our learned continuous-time variance-minimizing noise schedule SNR$(t)$ for CIFAR-10, compared to its log-linear initialization and to schedules from the literature: [1] The $\beta$-Linear schedule from Ho et al. [2020], [2] The $\alpha$-Cosine schedule from Nichol and Dhariwal [2021]. All schedules were scaled and shifted on the log scale such that the resulting SNR$_{\min}$, SNR$_{\max}$ were the equal to our learned endpoints, resulting in the same VLB estimate of 2.66. We report the variance of our VLB estimate per data point, computed on the test set, and conditional on the data: This does not include the noise due to sampling minibatches of data.

| $T_{train}$ | $T_{eval}$ | BPD | Bits-Back Net BPD |
|---|---|---|---|
| 10 | 10 | 4.31 | |
| 100 | 100 | 2.84 | |
| 250 | 250 | 2.73 | |
| 500 | 500 | 2.68 | |
| 1000 | 1000 | 2.67 | |
| 10000 | 10000 | 2.66 | |
| $\infty$ | 10 | 7.54 | 7.54 |
| $\infty$ | 100 | 2.90 | 2.91 |
| $\infty$ | 250 | 2.74 | 2.76 |
| $\infty$ | 500 | 2.69 | 2.72 |
| $\infty$ | 1000 | 2.67 | 2.72 |
| $\infty$ | 10000 | **2.65** | |
| $\infty$ | $\infty$ | **2.65** | |

Table 2: Discrete versus continuous-time training and evaluation with CIFAR-10, in terms of bits per dimension (BPD).

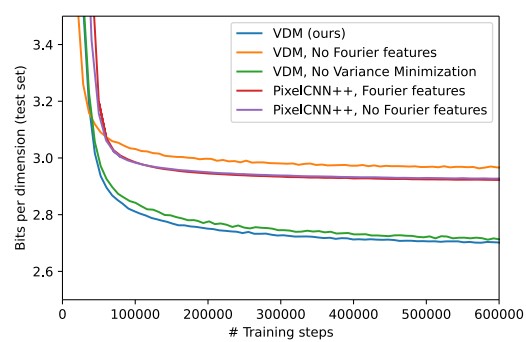

Figure 5: Test set likelihoods during training, with/without Fourier features, and with/without learning the noise schedule to minimize variance.

ANS [Townsend et al., 2018], an implementation of bits-back coding based on asymmetric numeral systems [Duda, 2009]. Details of our implementation are given in Appendix N. We achieve state-of-the-art net codelengths, proving our model can be used as the basis of a lossless compression algorithm. However, for large $T_{eval}$ a gap remains with the theoretically optimal codelength corresponding to the negative VLB, and compression becomes computationally expensive due to the large number of neural network forward passes required. Closing this gap with more efficient implementations of bits-back coding suitable for very deep models is an interesting avenue for future work.

## 7 Conclusion

We presented state-of-the-art results on modeling the density of natural images using a new class of diffusion models that incorporates a learnable diffusion specification, Fourier features for fine-scale modeling, as well as other architectural innovations. In addition, we obtained new theoretical insight into likelihood-based generative modeling with diffusion models, showing a surprising invariance of the VLB to the forward time diffusion process in continuous time, as well as an equivalence between various diffusion processes from the literature previously thought to be different.

## Acknowledgments

We thank Yang Song, Kevin Murphy and Mohammad Norouzi for feedback on drafts of this paper.

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
