# A Distribution details

## A.1 $q(\mathbf{z}_t|\mathbf{z}_s)$

The distribution of $\mathbf{z}_t$ given $\mathbf{z}_s$, for any $0 \le s < t \le 1$, is given by:

$$q(\mathbf{z}_t|\mathbf{z}_s) = \mathcal{N}\left(\alpha_{t|s}\mathbf{z}_s, \sigma_{t|s}^2\mathbf{I}\right), \tag{20}$$

where

$$\alpha_{t|s} = \alpha_t/\alpha_s, \tag{21}$$

and

$$\sigma_{t|s}^2 = \sigma_t^2 - \alpha_{t|s}^2\sigma_s^2. \tag{22}$$

## A.2 $q(\mathbf{z}_s|\mathbf{z}_t, \mathbf{x})$

Given the distributions $q(\mathbf{z}_t|\mathbf{x})$ (Equation 1) and $q(\mathbf{z}_t|\mathbf{z}_s)$ (Equation 22), we apply Bayes rule to derive the distribution $q(\mathbf{z}_s|\mathbf{z}_t, \mathbf{x})$ for any $0 \le s < t \le 1$, yielding:

$$q(\mathbf{z}_s|\mathbf{z}_t, \mathbf{x}) = \mathcal{N}(\boldsymbol{\mu}_Q(\mathbf{z}_t, \mathbf{x}; s, t), \sigma_Q^2(s, t)\mathbf{I}) \tag{23}$$

$$\text{where } \sigma_Q^2(s, t) = \sigma_{t|s}^2\sigma_s^2/\sigma_t^2, \tag{24}$$

$$\text{and } \boldsymbol{\mu}_Q(\mathbf{z}_t, \mathbf{x}; s, t) = \frac{1}{\alpha_{t|s}}(\mathbf{z}_t + \sigma_{t|s}^2\nabla_{\mathbf{z}_t}\log q(\mathbf{z}_t|\mathbf{x})) = \frac{\alpha_{t|s}\sigma_s^2}{\sigma_t^2}\mathbf{z}_t + \frac{\alpha_s\sigma_{t|s}^2}{\sigma_t^2}\mathbf{x}. \tag{25}$$

## A.3 $p(\mathbf{z}_s|\mathbf{z}_t)$

Finally, we choose the conditional model distributions as

$$p(\mathbf{z}_s|\mathbf{z}_t) = q(\mathbf{z}_s|\mathbf{z}_t, \mathbf{x} = \hat{\mathbf{x}}_{\boldsymbol{\theta}}(\mathbf{z}_t; t)), \tag{26}$$

i.e. the same as $q(\mathbf{z}_s|\mathbf{z}_t, \mathbf{x})$, but with the original data $\mathbf{x}$ replaced by the output of a denoising model $\hat{\mathbf{x}}_{\boldsymbol{\theta}}(\mathbf{z}_t; t)$ that predicts $\mathbf{x}$ from its noisy version $\mathbf{z}_t$. We then have

$$p(\mathbf{z}_s|\mathbf{z}_t) = \mathcal{N}(\mathbf{z}_s; \boldsymbol{\mu}_{\boldsymbol{\theta}}(\mathbf{z}_t; s, t), \sigma_Q^2(s, t)\mathbf{I}) \tag{27}$$

with variance $\sigma_Q^2(s, t)$ the same as in Equation 24, and

$$\begin{aligned}
\boldsymbol{\mu}_{\boldsymbol{\theta}}(\mathbf{z}_t; s, t) &= \frac{\alpha_{t|s}\sigma_s^2}{\sigma_t^2}\mathbf{z}_t + \frac{\alpha_s\sigma_{t|s}^2}{\sigma_t^2}\hat{\mathbf{x}}_{\boldsymbol{\theta}}(\mathbf{z}_t; t) \\
&= \frac{1}{\alpha_{t|s}}\mathbf{z}_t - \frac{\sigma_{t|s}^2}{\alpha_{t|s}\sigma_t}\hat{\boldsymbol{\epsilon}}_{\boldsymbol{\theta}}(\mathbf{z}_t; t) \\
&= \frac{1}{\alpha_{t|s}}\mathbf{z}_t + \frac{\sigma_{t|s}^2}{\alpha_{t|s}}\mathbf{s}_{\boldsymbol{\theta}}(\mathbf{z}_t; t),
\end{aligned} \tag{28}$$

where

$$\hat{\boldsymbol{\epsilon}}_{\boldsymbol{\theta}}(\mathbf{z}_t; t) = (\mathbf{z}_t - \alpha_t\hat{\mathbf{x}}_{\boldsymbol{\theta}}(\mathbf{z}_t; t))/\sigma_t \tag{29}$$

and

$$\mathbf{s}_{\boldsymbol{\theta}}(\mathbf{z}_t; t) = (\alpha_t\hat{\mathbf{x}}_{\boldsymbol{\theta}}(\mathbf{z}_t; t) - \mathbf{z}_t)/\sigma_t^2. \tag{30}$$

Equation 28 shows that we can interpret our model in three different ways:

1. In terms of the *denoising model* $\hat{\mathbf{x}}_{\boldsymbol{\theta}}(\mathbf{z}_t; t)$ that recovers $\mathbf{x}$ from its corrupted version $\mathbf{z}_t$.
2. In terms of a *noise prediction model* $\hat{\boldsymbol{\epsilon}}_{\boldsymbol{\theta}}(\mathbf{z}_t; t)$ that directly infers the noise $\boldsymbol{\epsilon}$ that was used to generate $\mathbf{z}_t$.
3. In terms of a *score model* $\mathbf{s}_{\boldsymbol{\theta}}(\mathbf{z}_t; t)$, that at its optimum equals the scores of the marginal density: $\mathbf{s}^*(\mathbf{z}_t; t) = \nabla_{\mathbf{z}}\log q(\mathbf{z}_t)$; see Appendix L.

These are three equally valid views on the same model class, that have been used interchangeably in the literature. We find the denoising interpretation the most intuitive, and will therefore mostly use $\hat{\mathbf{x}}_{\boldsymbol{\theta}}(\mathbf{z}_t; t)$ in the theoretical part of this paper, although in practice we parameterize our model via $\hat{\boldsymbol{\epsilon}}_{\boldsymbol{\theta}}(\mathbf{z}_t; t)$ following Ho et al. [2020]. The parameterization of our model is discussed in Appendix B.

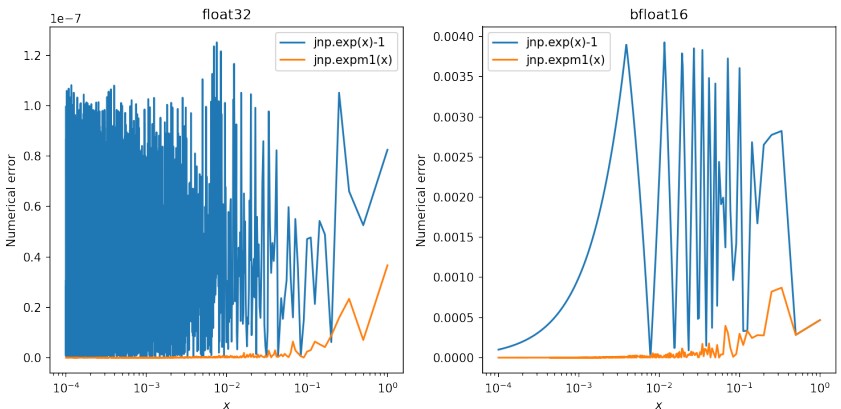

Figure 6: For optimization of the discrete-time diffusion loss and sampling from either the continuous-time or discrete-time model, an operation of the form $\exp(x)-1$ needs to be performed. This operation can result in large numerical errors when performed with 32-bit (float32) or 16-bit (e.g. bfloat16) floating point numbers. For this reason, many numerical packages implement the numerically more stable version $\text{expm1}(x) = \exp(x) - 1$. We here plot the absolute numerical errors for each of these versions for float32 and bfloat16. For this plot we used $\text{expm1}(.)$ with 64-bit floating point (float64) as ground truth. The plotted numerical error is the absolute value of the difference between the ground truth and the computed float32 or bfloat16 values.

## A.4 Further simplification of $p(\mathbf{z}_s|\mathbf{z}_t)$

After plugging in the specifications of $\sigma_t$, $\alpha_t$ and $\hat{\mathbf{x}}_{\boldsymbol{\theta}}(\mathbf{z}_t; t)$ that we use in experiments, given in Sections 3.2 and 3.4, it can be verified that the distribution $p(\mathbf{z}_s|\mathbf{z}_t) = \mathcal{N}(\boldsymbol{\mu}_{\boldsymbol{\theta}}(\mathbf{z}_t; s, t), \sigma_Q^2(s, t)\mathbf{I})$ simplifies to:

$$\boldsymbol{\mu}_{\boldsymbol{\theta}}(\mathbf{z}_t; s, t) = \frac{\alpha_s}{\alpha_t}\left(\mathbf{z}_t + \text{expm1}(\gamma_{\boldsymbol{\eta}}(s) - \gamma_{\boldsymbol{\eta}}(t))\hat{\boldsymbol{\epsilon}}_{\boldsymbol{\theta}}(\mathbf{z}_t; t)\right) \tag{31}$$

$$\sigma_Q^2(s, t) = \sigma_s^2 \cdot (-\text{expm1}(\gamma_{\boldsymbol{\eta}}(s) - \gamma_{\boldsymbol{\eta}}(t))) \tag{32}$$

where $\alpha_s = \sqrt{\text{sigmoid}(-\gamma_{\boldsymbol{\eta}}(s))}$, $\alpha_t = \sqrt{\text{sigmoid}(-\gamma_{\boldsymbol{\eta}}(t))}$, $\sigma_s^2 = \text{sigmoid}(\gamma_{\boldsymbol{\eta}}(s))$, and where $\text{expm1}(.) = \exp(.) - 1$; see Figure 6. Ancestral sampling from this distribution can be performed through simply doing:

$$\mathbf{z}_s = \sqrt{\alpha_s^2/\alpha_t^2}(\mathbf{z}_t - c\hat{\boldsymbol{\epsilon}}_{\boldsymbol{\theta}}(\mathbf{z}_t; t)) + \sqrt{(1 - \alpha_s^2)c}\boldsymbol{\epsilon} \tag{33}$$

where $\alpha_s^2 = \text{sigmoid}(-\gamma_{\boldsymbol{\eta}}(s))$, $\alpha_t^2 = \text{sigmoid}(-\gamma_{\boldsymbol{\eta}}(t))$, $c = -\text{expm1}(\gamma_{\boldsymbol{\eta}}(s) - \gamma_{\boldsymbol{\eta}}(t))$, and $\boldsymbol{\epsilon} \sim \mathcal{N}(0, \mathbf{I})$.

## B Hyperparameters, architecture, and implementation details

In this section we provide details on the exact setup for each of our experiments. In Sections B.1 we describe the choices in common to each of our experiments. Hyperparameters specific to the individual experiments are given in Section B.2. We are currently working towards open sourcing our code.

### B.1 Model and implementation

Our denoising models are parameterized in terms of noise prediction models $\hat{\boldsymbol{\epsilon}}_{\boldsymbol{\theta}}(\mathbf{z}_t; \gamma_t)$, as explained in Section 3.4. Our noise prediction models $\hat{\boldsymbol{\epsilon}}_{\boldsymbol{\theta}}$ closely follow the architecture used by Ho et al. [2020], which is based on a U-Net type neural net [Ronneberger et al., 2015] that maps from the input $\mathbf{z} \in \mathbb{R}^d$ to output of the same dimension. As compared to their publically available code at https://github.com/hojonathanho/diffusion, our implementation differs in the following ways:

- Our networks don't perform any internal downsampling or upsampling: we process all the data at the original input resolution.

- Our models are deeper than those used by Ho et al. [2020]. Specific numbers are given in Section B.2.

- Instead of taking time $t$ as input to the noise prediction model, we use $\gamma_t$, which we rescale to have approximately the same range as $t$ of $[0, 1]$ before using it to form 'time' embeddings in the same way as Ho et al. [2020].

- Our models calculate Fourier features on the input data $\mathbf{z}_t$ as discussed in Appendix C, which are then concatenated to $\mathbf{z}_t$ before being fed to the U-Net.

- Apart from the *middle* attention block that connects the upward and downward branches of the U-Net, we remove all other attention blocks from the model. We found that these attention blocks made it more likely for the model to overfit to the training set.

- All of our models use dropout at a rate of 0.1 in the intermediate layers, as did Ho et al. [2020]. In addition we regularize the model by using decoupled weight decay [Loshchilov and Hutter, 2017] with coefficient 0.01.

- We use the Adam optimizer with a learning rate of $2e^{-4}$ and exponential decay rates of $\beta_1 = 0.9, \beta_2 = 0.99$. We found that higher values for $\beta_2$ resulted in training instabilities.

- For evaluation, we use an exponential moving average of our parameters, calculated with an exponential decay rate of 0.9999.

We regularly evaluate the variational bound on the likelihood on the validation set and find that our models do not overfit during training, using the current settings. We therefore do not use early stopping and instead allow the network to be optimized for 10 million parameter updates for CIFAR-10, and for 2 million updates for ImageNet, before obtaining the test set numbers reported in this paper. It looks like our models keep improving even after this number of updates, in terms of likelihood, but we did not explore this systematically due to resource constraints.

All of our models are trained on TPUv3 hardware (see https://cloud.google.com/tpu) using data parallelism. We also evaluated our trained models using CPU and GPU to check for robustness of our reported numbers to possible rounding errors. We found only very small differences when evaluating on these other hardware platforms.

## B.2 Settings for each dataset

Our model for CIFAR-10 with no data augmentation uses a U-Net of depth 32, consisting of 32 ResNet blocks in the forward direction and 32 ResNet blocks in the reverse direction, with a single attention layer and two additional ResNet blocks in the middle. We keep the number of channels constant throughout at 128. This model was trained on 8 TPUv3 chips, with a total batch size of 128 examples. Reaching a test-set BPD of 2.65 after 10 million updates takes 9 days, although our model already surpasses sparse transformers (the previous state-of-the-art) of 2.80 BPD after only 2.5 hours of training.

For CIFAR-10 with data augmentation we used random flips, 90-degree rotations, and color channel swapping, which were previously shown to help for density estimation by Jun et al. [2020]. Each of the three augmentations independently were given a $50\%$ probability of being applied to each example, which means that 1 in 8 training examples was not augmented at all. For this experiment, we doubled the number of channels in our model to 256, and decreased the dropout rate from $10\%$ to $5\%$. Since overfitting was less of a problem with data augmentation, we add back the attention blocks after each ResNet block, following Ho et al. [2020]. We also experimented with conditioning our model on an additional binary feature that indicates whether or not the example was augmented, which can be seen as a simplified version of the augmentation conditioning proposed by Jun et al. [2020]. Conditioning made almost no difference to our results, which may be explained by the relatively large fraction ($12.5\%$) of clean data fed to our model during training. We trained our model for slightly over a week on 128 TPUv3 chips to obtain the reported result.

Our model for 32x32 ImageNet looks similar to that for CIFAR-10 without data augmentation, with a U-Net depth of 32, but uses double the number of channels at 256. It is trained using data parallelism on 32 TPUv3 chips, with a total batch size of 512.

Our model for 64x64 ImageNet uses double the depth at 64 ResNet layers in both the forward and backward direction in the U-Net. It also uses a constant number of channels of 256. This model is trained on 128 TPUv3 chips at a total batch size of 512 examples.

## C Fourier features for improved fine scale prediction

Prior work on diffusion models has mainly focused on the perceptual quality of generated samples, which emphasizes coarse scale patterns and global consistency of generated images. Here, we optimize for likelihood, which is sensitive to fine scale details and exact values of individual pixels. Since our reconstruction model $p(\mathbf{x}|\mathbf{z}_0)$ given in Equation 8 is weak, the burden of modeling these fine scale details falls on our denoising diffusion model $\hat{\mathbf{x}}_{\boldsymbol{\theta}}$. In initial experiments, we found that the denoising model had a hard time accurately modeling these details. At larger noise levels, the latents $\mathbf{z}_t$ follow a smooth distribution due to the added Gaussian noise, but at the smallest noise levels the discrete nature of 8-bit image data leads to sharply peaked marginal distributions $q(\mathbf{z}_t)$.

To capture the fine scale details of the data, we propose adding a set of *Fourier features* to the input of our denoising model $\hat{\mathbf{x}}_{\boldsymbol{\theta}}(\mathbf{z}_t; t)$. Such Fourier features consist of a linear projection of the original data onto a set of periodic basis functions with high frequency, which allows the network to more easily model high frequency details of the data. Previous work [Tancik et al., 2020] has used these features for input coordinates to model high frequency details across the *spatial* dimension, and for time embeddings to condition denoising networks over the *temporal* dimension [Song et al., 2021b]. Here we apply it to color channels for single pixels, in order to model fine distributional details at the level of each scalar input.

Concretely, let $z_{i,j,k}$ be the scalar value in the $k$-th channel in the $(i,j)$ spatial position of network input $\mathbf{z}_t$. We then add additional channels to the input of the denoising model of the form

$$f_{i,j,k}^n = \sin(z_{i,j,k}2^n\pi), \quad \text{and} \quad g_{i,j,k}^n = \cos(z_{i,j,k}2^n\pi), \tag{34}$$

where $n$ runs over a range of integers $\{n_{min}, ..., n_{max}\}$. These additional channels are then concatenated to $\mathbf{z}_t$ before being used as input in a standard convolutional denoising model similar to that used by Ho et al. [2020]. We find that the presence of these high frequency features allows our network to learn with much higher values of $\text{SNR}_{\max}$, or conversely lower noise levels $\sigma_0^2$, than is otherwise optimal. This leads to large improvements in likelihood as demonstrated in Section 6 and Figure 5. We did not observe such improvements when incorporating Fourier features into autoregressive models.

In our expreriments, we got best results with $n_{min} = 7$ and $n_{max} = 8$, probably since Fourier features with these frequencies are most relevant; features with lower frequencies can be learned by the network from $\mathbf{z}$, and higher frequencies are not present in the data thus irrelevant for likelihood.

## D As a SDE

When we take the number of steps $T \to \infty$, our model for $p(\mathbf{z}_t)$ can best be described as a continuous time diffusion process [Song et al., 2021b], governed by the stochastic differential equation

$$d\mathbf{z}_t = [f(t)\mathbf{z}_t - g^2(t)s_{\boldsymbol{\theta}}(\mathbf{z}_t; t)]dt + g(t)d\mathbf{W}_t, \tag{35}$$

with time running backwards from $t = 1$ to $t = 0$, where $\mathbf{W}$ denotes a Wiener process (standard Brownian motion), and

$$f(t) = \frac{d\log\alpha_t}{dt}, \quad g^2(t) = \frac{d\sigma^2(t)}{dt} - 2\frac{d\log\alpha_t}{dt}\sigma^2(t). \tag{36}$$

## E Derivation of the VLB estimators

### E.1 Discrete-time VLB

Similar to [Sohl-Dickstein et al., 2015], we decompose the negative variational lower bound (VLB) as:

$$-\log p(\mathbf{x}) \leq -\text{VLB}(\mathbf{x}) = \underbrace{D_{KL}(q(\mathbf{z}_1|\mathbf{x})||p(\mathbf{z}_1))}_{\text{Prior loss}} + \underbrace{\mathbb{E}_{q(\mathbf{z}_0|\mathbf{x})}[-\log p(\mathbf{x}|\mathbf{z}_0)]}_{\text{Reconstruction loss}} + \underbrace{\mathcal{L}_T(\mathbf{x})}_{\text{Diffusion loss}}. \tag{37}$$

The prior loss and reconstruction loss can be (stochastically and differentiably) estimated using standard techniques. We will now derive an estimator for the diffusion loss $\mathcal{L}_T(\mathbf{x})$, the remaining and more challenging term. In the case of finite $T$, using $s(i) = (i-1)/T$, $t(i) = i/T$, the diffusion loss is:

$$\mathcal{L}_T(\mathbf{x}) = \sum_{i=1}^{T} \mathbb{E}_{q(\mathbf{z}_{t(i)}|\mathbf{x})} D_{KL}[q(\mathbf{z}_{s(i)}|\mathbf{z}_{t(i)}, \mathbf{x})||p(\mathbf{z}_{s(i)}|\mathbf{z}_{t(i)})]. \tag{38}$$

We will use $s$ and $t$ as shorthands for $s(i)$ and $t(i)$. We will first derive an expression of $D_{KL}(q(\mathbf{z}_s|\mathbf{z}_t, \mathbf{x})||p(\mathbf{z}_s|\mathbf{z}_t))$.

Recall that $p(\mathbf{z}_s|\mathbf{z}_t) = q(\mathbf{z}_s|\mathbf{z}_t, \mathbf{x} = \hat{\mathbf{x}}_{\boldsymbol{\theta}}(\mathbf{z}_t; t))$, and thus $q(\mathbf{z}_s|\mathbf{z}_t, \mathbf{x}) = \mathcal{N}(\mathbf{z}_s; \boldsymbol{\mu}_Q(\mathbf{z}_t, \mathbf{x}; s, t), \sigma_Q^2(s, t)\mathbf{I})$ and $p(\mathbf{z}_s|\mathbf{z}_t) = \mathcal{N}(\mathbf{z}_s; \boldsymbol{\mu}_{\boldsymbol{\theta}}(\mathbf{z}_t; s, t), \sigma_Q^2(s, t)\mathbf{I})$, with

$$\boldsymbol{\mu}_Q(\mathbf{z}_t, \mathbf{x}; s, t) = \frac{\alpha_{t|s}\sigma_s^2}{\sigma_t^2}\mathbf{z}_t + \frac{\alpha_s\sigma_{t|s}^2}{\sigma_t^2}\mathbf{x} \tag{39}$$

$$\boldsymbol{\mu}_{\boldsymbol{\theta}}(\mathbf{z}_t; s, t) = \frac{\alpha_{t|s}\sigma_s^2}{\sigma_t^2}\mathbf{z}_t + \frac{\alpha_s\sigma_{t|s}^2}{\sigma_t^2}\hat{\mathbf{x}}_{\boldsymbol{\theta}}(\mathbf{z}_t; t), \tag{40}$$

$$\text{and } \sigma_Q^2(s, t) = \sigma_{t|s}^2\sigma_s^2/\sigma_t^2. \tag{41}$$

Since $q(\mathbf{z}_s|\mathbf{z}_t, \mathbf{x})$ and $p(\mathbf{z}_s|\mathbf{z}_t)$ are Gaussians, their KL divergence is available in closed form as a function of their means and variances, which due to their with equal variances simplifies as:

$$D_{KL}(q(\mathbf{z}_s|\mathbf{z}_t, \mathbf{x})||p(\mathbf{z}_s|\mathbf{z}_t)) = \frac{1}{2\sigma_Q^2(s, t)}||\boldsymbol{\mu}_Q - \boldsymbol{\mu}_{\boldsymbol{\theta}}||_2^2 \tag{42}$$

$$= \frac{\sigma_t^2}{2\sigma_{t|s}^2\sigma_s^2}\frac{\alpha_s^2\sigma_{t|s}^4}{\sigma_t^4}||\mathbf{x} - \hat{\mathbf{x}}_{\boldsymbol{\theta}}(\mathbf{z}_t; t)||_2^2 \tag{43}$$

$$= \frac{1}{2\sigma_s^2}\frac{\alpha_s^2\sigma_{t|s}^2}{\sigma_t^2}||\mathbf{x} - \hat{\mathbf{x}}_{\boldsymbol{\theta}}(\mathbf{z}_t; t)||_2^2 \tag{44}$$

$$= \frac{1}{2\sigma_s^2}\frac{\alpha_s^2(\sigma_t^2 - \alpha_{t|s}^2\sigma_s^2)}{\sigma_t^2}||\mathbf{x} - \hat{\mathbf{x}}_{\boldsymbol{\theta}}(\mathbf{z}_t; t)||_2^2 \tag{45}$$

$$= \frac{1}{2}\frac{\alpha_s^2\sigma_t^2/\sigma_s^2 - \alpha_t^2}{\sigma_t^2}||\mathbf{x} - \hat{\mathbf{x}}_{\boldsymbol{\theta}}(\mathbf{z}_t; t)||_2^2 \tag{46}$$

$$= \frac{1}{2}\left(\frac{\alpha_s^2}{\sigma_s^2} - \frac{\alpha_t^2}{\sigma_t^2}\right)||\mathbf{x} - \hat{\mathbf{x}}_{\boldsymbol{\theta}}(\mathbf{z}_t; t)||_2^2 \tag{47}$$

$$= \frac{1}{2}\left(\text{SNR}(s) - \text{SNR}(t)\right)||\mathbf{x} - \hat{\mathbf{x}}_{\boldsymbol{\theta}}(\mathbf{z}_t; t)||_2^2 \tag{48}$$

Reparameterizing $\mathbf{z}_t \sim q(\mathbf{z}_t|\mathbf{x})$ as $\mathbf{z}_t = \alpha_t\mathbf{x} + \sigma_t\boldsymbol{\epsilon}$, where $\boldsymbol{\epsilon} \sim \mathcal{N}(0, \mathbf{I})$, our diffusion loss becomes:

$$\mathcal{L}_T(\mathbf{x}) = \sum_{i=1}^{T} \mathbb{E}_{q(\mathbf{z}_t|\mathbf{x})}[D_{KL}(q(\mathbf{z}_s|\mathbf{z}_t, \mathbf{x})||p(\mathbf{z}_s|\mathbf{z}_t))] \tag{49}$$

$$= \frac{1}{2}\mathbb{E}_{\boldsymbol{\epsilon}\sim\mathcal{N}(0,\mathbf{I})}[\sum_{i=1}^{T}(\text{SNR}(s) - \text{SNR}(t))||\mathbf{x} - \hat{\mathbf{x}}_{\boldsymbol{\theta}}(\mathbf{z}_t; t)||_2^2] \tag{50}$$

### E.2   Estimator of $\mathcal{L}_T(\mathbf{x})$

To avoid having to compute all $T$ terms when calculating the diffusion loss, we construct an unbiased estimator of $\mathcal{L}_T(\mathbf{x})$ using

$$\mathcal{L}_T(\mathbf{x}) = \frac{T}{2}\mathbb{E}_{\boldsymbol{\epsilon}\sim\mathcal{N}(0,\mathbf{I}), i\sim U\{1,T\}}\left[(\text{SNR}(s) - \text{SNR}(t))||\mathbf{x} - \hat{\mathbf{x}}_{\boldsymbol{\theta}}(\mathbf{z}_t; t)||_2^2\right] \tag{51}$$

where $U\{1, T\}$ is the discrete uniform distribution from 1 to (and including) $T$, $s = (i-1)/T$, $t = i/T$ and $\mathbf{z}_t = \alpha_t\mathbf{x} + \sigma_t\boldsymbol{\epsilon}$. This trivially yields an unbiased Monte Carlo estimator, by drawing random samples $i \sim U\{1, T\}$ and $\boldsymbol{\epsilon} \sim \mathcal{N}(0, \mathbf{I})$.

### E.3 Infinite depth ($T \to \infty$)

To calculate the limit of the diffusion loss as $T \to \infty$, we express $\mathcal{L}_T(\mathbf{x})$ as a function of $\tau = 1/T$:

$$\mathcal{L}_T(\mathbf{x}) = \frac{1}{2}\mathbb{E}_{\boldsymbol{\epsilon} \sim \mathcal{N}(0,\mathbf{I}), i \sim U\{1,T\}} \left[ \frac{\text{SNR}(t-\tau) - \text{SNR}(t)}{\tau} ||\mathbf{x} - \hat{\mathbf{x}}_{\boldsymbol{\theta}}(\mathbf{z}_t; t)||_2^2 \right], \quad (52)$$

where again $t = i/T$ and $\mathbf{z}_t = \alpha_t \mathbf{x} + \sigma_t \boldsymbol{\epsilon}$.

As $\tau \to 0, T \to \infty$, and letting $\text{SNR}'(t)$ denote the derivative of the SNR function, this then gives

$$\mathcal{L}_\infty(\mathbf{x}) = -\frac{1}{2}\mathbb{E}_{\boldsymbol{\epsilon} \sim \mathcal{N}(0,\mathbf{I}), t \sim U[0,1]} \left[ \text{SNR}'(t)||\mathbf{x} - \hat{\mathbf{x}}_{\boldsymbol{\theta}}(\mathbf{z}_t; t)||_2^2 \right] \quad (53)$$

$$= -\frac{1}{2}\mathbb{E}_{\boldsymbol{\epsilon} \sim \mathcal{N}(0,\mathbf{I})} \int_0^1 \text{SNR}'(t) \, ||\mathbf{x} - \hat{\mathbf{x}}_{\boldsymbol{\theta}}(\mathbf{z}_t; t)||_2^2 \, dt. \quad (54)$$

## F  Influence of the number of steps $T$ on the VLB

Recall that the diffusion loss for our choice of model $p, q$, when using $T$ timesteps, is given by

$$\mathcal{L}_T(\mathbf{x}) = \frac{1}{2}\mathbb{E}_{\boldsymbol{\epsilon} \sim \mathcal{N}(0,\mathbf{I})} \left[ \sum_{i=1}^{T} \left( \text{SNR}(s(i)) - \text{SNR}(t(i)) \right) ||\mathbf{x} - \hat{\mathbf{x}}_{\boldsymbol{\theta}}(\mathbf{z}_{t(i)}; t(i))||_2^2 \right], \quad (55)$$

with $s(i) = (i-1)/T, t(i) = i/T$.

This can then be written equivalently as

$$\mathcal{L}_T(\mathbf{x}) = \frac{1}{2}\mathbb{E}_{\boldsymbol{\epsilon} \sim \mathcal{N}(0,\mathbf{I})} \left[ \sum_{i=1}^{T} \left( \text{SNR}(s) - \text{SNR}(t') + \text{SNR}(t') - \text{SNR}(t) \right) ||\mathbf{x} - \hat{\mathbf{x}}_{\boldsymbol{\theta}}(\mathbf{z}_t; t)||_2^2 \right], \quad (56)$$

with $t' = t - 0.5/T$.

In contrast, the diffusion loss with 2T timesteps can be written as

$$\mathcal{L}_{2T}(\mathbf{x}) = \frac{1}{2}\mathbb{E}_{\boldsymbol{\epsilon} \sim \mathcal{N}(0,\mathbf{I})} \sum_{i=1}^{T} \left( \text{SNR}(s) - \text{SNR}(t') \right) ||\mathbf{x} - \hat{\mathbf{x}}_{\boldsymbol{\theta}}(\mathbf{z}_{t'}; t')||_2^2$$

$$+ \sum_{i=1}^{T} \left( \text{SNR}(t') - \text{SNR}(t) \right) ||\mathbf{x} - \hat{\mathbf{x}}_{\boldsymbol{\theta}}(\mathbf{z}_t; t)||_2^2. \quad (57)$$

Subtracting the two results, we get

$$\mathcal{L}_{2T}(\mathbf{x}) - \mathcal{L}_T(\mathbf{x}) =$$
$$\frac{1}{2}\mathbb{E}_{\boldsymbol{\epsilon} \sim \mathcal{N}(0,\mathbf{I})} \left[ \sum_{i=1}^{T} \left( \text{SNR}(s) - \text{SNR}(t') \right) \left( ||\mathbf{x} - \hat{\mathbf{x}}_{\boldsymbol{\theta}}(\mathbf{z}_{t'}; t')||_2^2 - ||\mathbf{x} - \hat{\mathbf{x}}_{\boldsymbol{\theta}}(\mathbf{z}_t; t)||_2^2 \right) \right]. \quad (58)$$

Since $t' < t$, $\mathbf{z}_{t'}$ is a less noisy version of the data from earlier in the diffusion process compared to $\mathbf{z}_t$. Predicting the original data $\mathbf{x}$ from $\mathbf{z}_{t'}$ is thus strictly easier than from $\mathbf{z}_t$, leading to lower mean squared error if our model $\hat{\mathbf{x}}_{\boldsymbol{\theta}}$ is good enough. We thus have that $\mathcal{L}_{2T}(\mathbf{x}) - \mathcal{L}_T(\mathbf{x}) < 0$, which means that doubling the number of timesteps always improves our diffusion loss. For this reason we argue for using the continuous-time VLB corresponding to $T \to \infty$ in this paper.

## G  Equivalence of diffusion specifications

(Continuation of Section 5.1.)

Let $p^A(\mathbf{x})$ denote the distribution on observed data $\mathbf{x}$ as defined by the combination of a diffusion specification $\{\alpha_v^A, \sigma_v^A\}$ and denoising function $\tilde{\mathbf{x}}_{\boldsymbol{\theta}}^A$, and let $\mathbf{z}_v^A$ denote the latents of this diffusion process. Similarly, let $p^B(\mathbf{x}), \mathbf{z}_v^B$ be defined through a different specification $\{\alpha_v^B, \sigma_v^B, \tilde{\mathbf{x}}_{\boldsymbol{\theta}}^B\}$.

Since $v \equiv \alpha_v^2/\sigma_v^2$, we have that $\sigma_v = \alpha_v/\sqrt{v}$, which means that $\mathbf{z}_v(\mathbf{x}, \boldsymbol{\epsilon}) = \alpha_v \mathbf{x} + \sigma_v \boldsymbol{\epsilon} = \alpha_v(\mathbf{x} + \boldsymbol{\epsilon}/\sqrt{v})$. This holds for any diffusion specification by definition, and therefore we have $\mathbf{z}_v^A(\mathbf{x}, \boldsymbol{\epsilon}) = (\alpha_v^A/\alpha_v^B)\mathbf{z}_v^B(\mathbf{x}, \boldsymbol{\epsilon})$. The latents $\mathbf{z}_v$ for different diffusion specifications are thus identical, up to a trivial rescaling, and their information content only depends on the signal-to-noise ratio $v$, not on $\alpha_v, \sigma_v$ separately.

For the purpose of denoising from a latent $\mathbf{z}_v^B$, this means we can simply define the denoising model as $\tilde{\mathbf{x}}_{\boldsymbol{\theta}}^B(\mathbf{z}_v^B, v) \equiv \tilde{\mathbf{x}}_{\boldsymbol{\theta}}^A((\alpha_v^A/\alpha_v^B)\mathbf{z}_v^B, v)$, and we will then get the same reconstruction loss as when denoising from $\mathbf{z}_v^A$ using model $\tilde{\mathbf{x}}_{\boldsymbol{\theta}}^A$. Assuming endpoints $\mathrm{SNR}_{\max}$ and $\mathrm{SNR}_{\min}$ are equal for both specifications, Equation 18 then tells us that $\mathcal{L}_{\infty}^A(\mathbf{x}) = \mathcal{L}_{\infty}^B(\mathbf{x})$, i.e. they both produce the same diffusion loss in continuous time.

Similarly, the conditional model distributions over the latents $\mathbf{z}_v$ in our generative model are functions of the denoising model $\tilde{\mathbf{x}}_{\boldsymbol{\theta}}(\mathbf{z}_v, v)$ (see Equation 26), and we therefore have that the specification $\{\alpha_v^A, \sigma_v^A, \tilde{\mathbf{x}}_{\boldsymbol{\theta}}^A\}$ defines the same generative model over $\mathbf{z}_v, \mathbf{x}$ as the specification $\{\alpha_v^B, \sigma_v^B, \tilde{\mathbf{x}}_{\boldsymbol{\theta}}^B\}$, again up to a rescaling of the latents by $\alpha_v^A/\alpha_v^B$.

# H   Implementation of monotonic neural net noise schedule $\gamma_{\boldsymbol{\eta}}(t)$

To learn the signal-to-noise ratio $\mathrm{SNR}(t)$, we parameterize it as $\mathrm{SNR}(t) = \exp(-\gamma_{\boldsymbol{\eta}}(t))$ with $\gamma_{\boldsymbol{\eta}}(t)$ a monotonic neural network. This network consists of 3 linear layers with weights that are restricted to be positive, $l_1, l_2, l_3$, which are composed as $\tilde{\gamma}_{\boldsymbol{\eta}}(t) = l_1(t) + l_3(\phi(l_2(l_1(t))))$, with $\phi$ the sigmoid function. Here, the $l_2$ layer has 1024 outputs, and the other layers have a single output.

In case of the continuous-time model, for the purpose of variance minimization, we postprocess the noise schedule as detailed in Appendix I.

# I   Variance minimization

Reduction of the variance diffusion loss estimator can lead to faster optimization.

For the continuous-time model, we reduce the variance of the diffusion loss estimator through two methods: (1) optimizing the noise schedule w.r.t. the variance of the diffusion loss, and (2) using a low-discrepency sampler. Note that these methods can be omitted if one aims for a simple implementation of our methods, at the expense of slower optimization.

## I.1   Low-discrepency sampler

When processing a minibatch of $k$ examples $\mathbf{x}^i, i \in \{1, \ldots, k\}$, we require $k$ timesteps $t^i$ sampled from a uniform distribution. Instead of sampling these timesteps independently, we sample a single uniform random number $u_0 \sim U[0,1]$ and then set $t^i = \mathrm{mod}(u_0 + i/k, 1)$. Each $t^i$ now has the correct uniform marginal distribution, but the minibatch of timesteps covers the space in $[0, 1]$ more equally than when sampling independently, which we find to reduce the variance in our VLB estimate.

## I.2   Optimizing the noise schedule w.r.t. the variance of the diffusion loss

In case of the continuous-time model, for the purpose of variance minimization, we postprocess the noise schedule as follows. At this point, the range of $\tilde{\gamma}_{\boldsymbol{\eta}}(t)$ is unbounded and so the resulting SNR is not yet restricted to $[\mathrm{SNR}_{\min}, \mathrm{SNR}_{\max}]$. We therefore postprocess the monotonic neural network as

$$\gamma_{\boldsymbol{\eta}}(t) = \gamma_0 + (\gamma_1 - \gamma_0)\frac{\tilde{\gamma}_{\boldsymbol{\eta}}(t) - \tilde{\gamma}_{\boldsymbol{\eta}}(0)}{\tilde{\gamma}_{\boldsymbol{\eta}}(1) - \tilde{\gamma}_{\boldsymbol{\eta}}(0)}, \tag{59}$$

with $\gamma_0 = -\log(\mathrm{SNR}_{\max}), \gamma_1 = -\log(\mathrm{SNR}_{\min})$. Now $\mathrm{SNR}(t) = \exp(-\gamma_{\boldsymbol{\eta}}(t))$ has the correct range and interpolates exactly between $\mathrm{SNR}_{\min}$ and $\mathrm{SNR}_{\max}$. We treat $\gamma_0, \gamma_1$ as free parameters that we optimize directly w.r.t. the VLB. The remaining parameters $\boldsymbol{\eta}$ are instead learned by minimizing the variance of the stochastic estimate of the VLB.

We minimize the variance by performing stochastic gradient descent on our squared diffusion loss $\mathcal{L}_{\infty}^{MC}(\mathbf{x}, w, \gamma)^2$. We have that $\mathbb{E}_{t, \boldsymbol{\epsilon}}[\mathcal{L}_{\infty}^{MC}(\mathbf{x}, w, \gamma)^2] = \mathcal{L}_{\infty}(\mathbf{x}, w)^2 + \mathrm{Var}_{t, \boldsymbol{\epsilon}}[\mathcal{L}_{\infty}^{MC}(\mathbf{x}, w, \gamma)]$, where

the first part is independent of $\gamma_{\boldsymbol{\eta}}(t)$, and hence that

$$\mathbb{E}_{t,\boldsymbol{\epsilon}}[\nabla_{\boldsymbol{\eta}}\mathcal{L}_{\infty}^{MC}(\mathbf{x},w,\gamma_{\boldsymbol{\eta}})^2] = \nabla_{\boldsymbol{\eta}}\text{Var}_{t,\boldsymbol{\epsilon}}[\mathcal{L}_{\infty}^{MC}(\mathbf{x},w,\gamma_{\boldsymbol{\eta}})]. \tag{60}$$

We can calculate this gradient with negligible computational overhead as a by-product of calculating the gradient of the VLB, details of which are given in Appendix H.

We wish to calculate $\nabla_{\boldsymbol{\eta}}[\mathcal{L}_{\infty}^{MC}(\mathbf{x},\gamma_{\boldsymbol{\eta}})^2]$ without performing a second backpropagation pass through the denoising model due to this objective being different than for the other parameters. To do this, we decompose the gradient as

$$\frac{d}{d\boldsymbol{\eta}}[\mathcal{L}_{\infty}^{MC}(\mathbf{x},\gamma_{\boldsymbol{\eta}})^2] = \frac{d}{d\text{SNR}}\left[\mathcal{L}_{\infty}^{MC}(\mathbf{x},\text{SNR})^2\right]\frac{d}{d\boldsymbol{\eta}}\left[\text{SNR}(\boldsymbol{\eta})\right], \tag{61}$$

$$\text{and } \frac{d}{d\text{SNR}}\left[\mathcal{L}_{\infty}^{MC}(\mathbf{x},\text{SNR})^2\right] = 2\frac{d}{d\text{SNR}}\left[\mathcal{L}_{\infty}^{MC}(\mathbf{x},\text{SNR})\right]\odot\mathcal{L}_{\infty}^{MC}(\mathbf{x},\text{SNR}), \tag{62}$$

where $\odot$ denotes elementwise multiplication. Here $\frac{d}{d\text{SNR}}\left[\mathcal{L}_{\infty}^{MC}(\mathbf{x},\text{SNR})\right]$ is computed along with the other gradients when performing the single backpropagation pass for calculating $\nabla_{\boldsymbol{\theta}}[\mathcal{L}_{\infty}^{MC}]$. The remaining operations required to get $\nabla_{\boldsymbol{\eta}}[\mathcal{L}_{\infty}^{MC}(\mathbf{x},\gamma_{\boldsymbol{\eta}})^2]$ have negligible computational cost.

This strategy of minimizing the variance of our diffusion loss estimate remains valid for weighted diffusion losses, $w(v) \neq 1$, not corresponding to the VLB, and we therefore expect it to be useful beyond the goal of optimizing for likelihood that we consider in this paper.

## J   Numerical stability

Floating point numbers are much worse at representing numbers close to 1, than at representing numbers close to 0. Since a naïve implementation of our model and its discrete-time loss function requires computing intermediate values that are close to 1, those numbers are erroneously rounded to 1, leading to numerical issues and incorrect results. Note that previous implementations of discrete-time diffusion models (e.g. [Ho et al., 2020]) used 64-bit floating point numbers to avoid numerical problems. We found this unnecessary in our model.

A numerically problematic term, for example, is $\boldsymbol{\sigma}_{t|s}^2$ which is used for sampling. It is straightforward to verify that:

$$\boldsymbol{\sigma}_{t|s}^2 = -\text{expm1}(\text{softplus}(\gamma(s)) - \text{softplus}(\gamma(t))), \tag{63}$$

where $\text{expm1}(x) \equiv \exp(x) - 1$ and $\text{softplus}(x) \equiv \log(1 + \exp(x))$ are functions with numerically stable primitives in common numerical computing packages.

## K   Comparison to DDPM and NCSN objectives

Previous works using denoising diffusion models [Ho et al., 2020, Song and Ermon, 2019, Nichol and Dhariwal, 2021] used a training objective that can be understood as a *weighted* diffusion loss of the form given in Equation 19:

$$\mathcal{L}_{\infty}(\mathbf{x},w) = \frac{1}{2}\mathbb{E}_{\boldsymbol{\epsilon}\sim\mathcal{N}(0,\mathbf{I})}\int_{\text{SNR}_{\min}}^{\text{SNR}_{\max}} w(v)\left\|\mathbf{x} - \tilde{\mathbf{x}}_{\boldsymbol{\theta}}(\mathbf{z}_v,v)\right\|_2^2 dv \tag{64}$$

$$= -\frac{1}{2}\mathbb{E}_{\boldsymbol{\epsilon}\sim\mathcal{N}(0,\mathbf{I})}\int_0^1 \text{SNR}'(t)w(\text{SNR}(t))\left\|\mathbf{x} - \hat{\mathbf{x}}_{\boldsymbol{\theta}}(\mathbf{z}_t;t)\right\|_2^2 dt \tag{65}$$

$$= \frac{1}{2}\mathbb{E}_{\boldsymbol{\epsilon}\sim\mathcal{N}(0,\mathbf{I})}\int_0^1 \gamma'(t)w(\exp(-\gamma(t)))\left\|\boldsymbol{\epsilon} - \hat{\boldsymbol{\epsilon}}_{\boldsymbol{\theta}}(\mathbf{z}_t;t)\right\|_2^2 dt, \tag{66}$$

where $\gamma(t) = -\log\text{SNR}(t)$.

When using the loss in Equation 64, we set $w(v) = 1$, corresponding to optimization of a variational bound on the likelihood of the data. Ho et al. [2020], Song and Ermon [2019], Nichol and Dhariwal [2021] instead choose to minimize the *simple objective* defined as

$$L_{\text{simple}}(\mathbf{x}) \equiv \int_0^1 \left\|\boldsymbol{\epsilon} - \hat{\boldsymbol{\epsilon}}_{\boldsymbol{\theta}}(\mathbf{z}_t,t)\right\|_2^2 dt, \tag{67}$$

or a discrete-time version of this.

Comparing Equation 67 with Equation 66, we can see that the loss used by Ho et al. [2020], Song and Ermon [2019], Nichol and Dhariwal [2021] corresponds to a weighting function $w(\text{SNR}(t)) = 1/\gamma'(t)$. Below, we derive the $\gamma(t)$, and thus the weighting function $w(v)$, corresponding to the diffusion processes used by Ho et al. [2020], Song and Ermon [2019], Nichol and Dhariwal [2021]. We visualize these weighting functions in Figure 7.

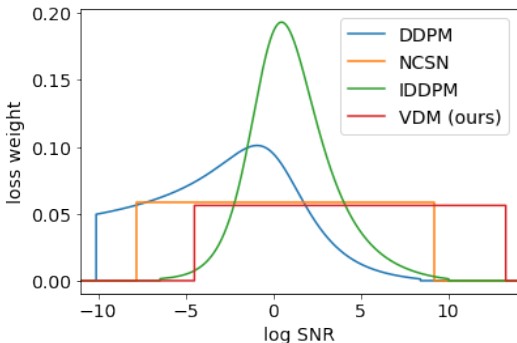

Figure 7: Implied weighting functions corresponding to the losses used by Ho et al. [2020], Song et al. [2020], and Nichol and Dhariwal [2021], as well as our proposed loss. NCSN [Song et al., 2020] uses a constant implied weighting function, and is thus consistent with maximization of the variational bound like we propose in this paper. However, unlike Song and Ermon [2019] we also learn the endpoints $\text{SNR}_{\min}, \text{SNR}_{\max}$, which results in a better optimized VLB value. DDPM [Ho et al., 2020] and improved DDPM [Nichol and Dhariwal, 2021] instead use implied weighting functions that put relatively more weight on the noisy data with low to medium signal-to-noise ratio. The latter two works report better FID and Inception Score than Song et al. [2020] and the current paper, which we hypothesize is due to their loss emphasizing the global consistence and coarse level patterns more than the fine scale features of the data.

For DDPM, Ho et al. [2020] use a diffusion process in discrete time with $\alpha_i = \sqrt{\prod_{j=1}^{i}(1-\beta_j)}$, $\sigma_i^2 = 1-\alpha_i^2$, where $\beta_i$ linearly interpolates between $\beta_1 = 1e^{-4}$ and $\beta_T = 0.02$ in $T = 1000$ discrete steps. When defining time $t = i/T$, this can be closely approximated as $\alpha_t^2 = \exp(-1e^{-4} - 10t^2)$, and correspondingly with $\text{SNR}(t) = 1/\text{expm1}(1e^{-4} + 10t^2)$ or $\gamma(t) = \log[\text{expm1}(1e^{-4} + 10t^2)]$, where $\text{expm1}(x) = \exp(x) - 1$. This approximation is shown in Figure 8.

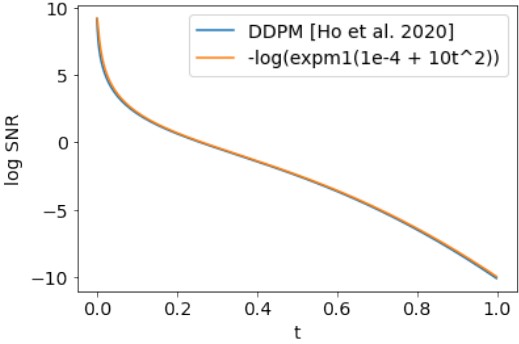

Figure 8: Log signal-to-noise ratio for the discrete-time diffusion process in Ho et al. [2020] and our continous-time approximation.

For NCSNv2, Song and Ermon [2020] instead use $\alpha_t = 1$ and let $\sigma_t$ be a geometric series interpolating between 0.01 and 50, i.e. $\sigma_t^2 = \exp(\gamma(t))$ with $\gamma(t) = 2\log[0.01] + 2\log[5000]t$. This means that $\gamma'(t) = 2\log[5000]$ and thus that $w(v)$ is a constant. The procedure proposed by Song and Ermon

[2020] is thus consistent with maximization of the VLB like we propose here. The same holds for [Song and Ermon, 2019].

For IDDPM, Nichol and Dhariwal [2021] use $\tilde{\alpha}_t = \cos(\frac{t+0.008}{1.008} \frac{\pi}{2}) / \cos(\frac{0.008}{1.008} \frac{\pi}{2})$. The values for $\tilde{\alpha}_t$ are then translated into value for $\beta_t$, which are then clipped to $0.999$. Subsequently we can then derive the $\alpha_t, \sigma_t, \gamma(t)$ corresponding to those $\beta_t$. Due to the clipping these expressions do not simplify, but we include their numerical results in Figure 7.

## L    Consistency

Let $q(\mathbf{x})$ denote the marginal distribution of data $\mathbf{x}$, and let:

$$q(\mathbf{z}_t) = \int q(\mathbf{z}_t|\mathbf{x})q(\mathbf{x})d\mathbf{x} \tag{68}$$

Here we will show that derived estimators are *consistent* estimators, in the sense that with infinite data, the optimal score model $\mathbf{s}_{\boldsymbol{\theta}}^*(\mathbf{z}_t; t)$ is such that:

$$\mathbf{s}_{\boldsymbol{\theta}}^*(\mathbf{z}_t; t) = \nabla_{\mathbf{z}} \log q(\mathbf{z}_t) \tag{69}$$

Note that $\nabla_{\mathbf{z}_t} \log q(\mathbf{z}_t|\mathbf{x}) = -\boldsymbol{\epsilon}/\boldsymbol{\sigma}_t$. We can rewrite the diffusion loss (discrete time or continuous time) for timestep $t$ as:

$$\mathcal{L}_T(\mathbf{x}; t) = \frac{1}{2} \mathbb{E}_{q(\mathbf{x}, \mathbf{z}_t)} \left[ \left\| \sqrt{\mathbf{c}(t)} \left( \nabla_{\mathbf{z}_t} \log q(\mathbf{z}_t|\mathbf{x}) - \mathbf{s}_{\boldsymbol{\theta}}(\mathbf{z}_t; t) \right) \right\|_2^2 \right] \tag{70}$$

where $\sqrt{\mathbf{c}(t)}$ is a time-dependent weighting factor.

In [Ho et al., 2020], it is noted that the discrete-time VLB, when using equal variances across dimensions, is equivalent to a Denoising Score Matching (DSM) objective [Vincent, 2011]. This is interesting, since it implies consistency. We generalize this original consistency proof of DSM to a more general case of different noises schedules per dimension, and arbitrary multipliers $\sqrt{\mathbf{c}_1}$ and $\sqrt{\mathbf{c}_2}$ in front of the scores, i.e. where the dimensions of $\mathbf{z}$ are differently weighted. Note, however that we'll only need the special case where $\sqrt{\mathbf{c}} = \sqrt{\mathbf{c}_1} = \sqrt{\mathbf{c}_2}$. First, note that:

$$\frac{1}{2}\mathbb{E}_{q(\mathbf{z}_t)} \left[ ||\sqrt{\mathbf{c}_1}\nabla_{\mathbf{z}_t} \log q(\mathbf{z}_t) - \sqrt{\mathbf{c}_2}\mathbf{s}_{\boldsymbol{\theta}}(\mathbf{z}_t)||_2^2 \right] = \frac{1}{2}\mathbb{E}_{q(\mathbf{z}_t)} \left[ ||\sqrt{\mathbf{c}_1}\nabla_{\mathbf{z}_t} \log q(\mathbf{z}_t)||_2^2 \right]$$
$$+ \frac{1}{2}\mathbb{E}_{q(\mathbf{z}_t)} \left[ ||\sqrt{\mathbf{c}_2}\mathbf{s}_{\boldsymbol{\theta}}(\mathbf{z}_t)||_2^2 \right]$$
$$- \mathbb{E}_{q(\mathbf{z}_t)} \left[ \langle \sqrt{\mathbf{c}_1}\nabla_{\mathbf{z}_t} \log q(\mathbf{z}_t), \sqrt{\mathbf{c}_2}\mathbf{s}_{\boldsymbol{\theta}}(\mathbf{z}_t) \rangle \right] \tag{71}$$

Where $\langle ., . \rangle$ denotes a dot product. Similarly:

$$\frac{1}{2}\mathbb{E}_{q(\mathbf{z}_t|\mathbf{x})} \left[ ||\sqrt{\mathbf{c}_1}\nabla_{\mathbf{z}_t} \log q(\mathbf{z}_t|\mathbf{x}) - \sqrt{\mathbf{c}_2}\mathbf{s}_{\boldsymbol{\theta}}(\mathbf{z}_t)||_2^2 \right] = \frac{1}{2}\mathbb{E}_{q(\mathbf{z}_t|\mathbf{x})} \left[ ||\sqrt{\mathbf{c}_1}\nabla_{\mathbf{z}_t} \log q(\mathbf{z}_t|\mathbf{x})||_2^2 \right]$$
$$+ \frac{1}{2}\mathbb{E}_{q(\mathbf{z}_t|\mathbf{x})} \left[ ||\sqrt{\mathbf{c}_2}\mathbf{s}_{\boldsymbol{\theta}}(\mathbf{z}_t)||_2^2 \right]$$
$$- \mathbb{E}_{q(\mathbf{z}_t|\mathbf{x})} \left[ \langle \sqrt{\mathbf{c}_1}\nabla_{\mathbf{z}_t} \log q(\mathbf{z}_t|\mathbf{x}), \sqrt{\mathbf{c}_2}\mathbf{s}_{\boldsymbol{\theta}}(\mathbf{z}_t) \rangle \right]$$
$$\tag{72}$$

The second terms of the right-hand sides of Equation 71 and Equation 72 are equal. The third terms of the right-hand sides of Equation 71 and Equation 72 are also equal:

$$\mathbb{E}_{q(\mathbf{z}_t)} \left[ \langle \sqrt{\mathbf{c_2}} \mathbf{s_\theta}(\mathbf{z}_t), \sqrt{\mathbf{c_1}} \nabla_{\mathbf{z}_t} \log q(\mathbf{z}_t) \rangle \right] \tag{73}$$

$$= \mathbb{E}_{q(\mathbf{z}_t)} \left[ \left\langle \sqrt{\mathbf{c_2}} \mathbf{s_\theta}(\mathbf{z}_t), \sqrt{\mathbf{c_1}} \frac{\nabla_{\mathbf{z}_t} q(\mathbf{z}_t)}{q(\mathbf{z}_t)} \right\rangle \right] \tag{74}$$

$$= \int_{\mathbf{z}_t} \langle \sqrt{\mathbf{c_2}} \mathbf{s_\theta}(\mathbf{z}_t), \sqrt{\mathbf{c_1}} \nabla_{\mathbf{z}_t} q(\mathbf{z}_t) \rangle \, d\mathbf{z}_t \tag{75}$$

$$= \int_{\mathbf{z}_t} \left\langle \sqrt{\mathbf{c_2}} \mathbf{s_\theta}(\mathbf{z}_t), \sqrt{\mathbf{c_1}} \nabla_{\mathbf{z}_t} \int_{\mathbf{x}} q(\mathbf{x}) q(\mathbf{z}_t|\mathbf{x}) d\mathbf{x} \right\rangle \, d\mathbf{z}_t \tag{76}$$

$$= \int_{\mathbf{z}_t} \left\langle \sqrt{\mathbf{c_2}} \mathbf{s_\theta}(\mathbf{z}_t), \sqrt{\mathbf{c_1}} \int_{\mathbf{x}} q(\mathbf{x}) \nabla_{\mathbf{z}_t} q(\mathbf{z}_t|\mathbf{x}) d\mathbf{x} \right\rangle \, d\mathbf{z}_t \tag{77}$$

$$= \int_{\mathbf{z}_t} \left\langle \sqrt{\mathbf{c_2}} \mathbf{s_\theta}(\mathbf{z}_t), \sqrt{\mathbf{c_1}} \int_{\mathbf{x}} q(\mathbf{x}) q(\mathbf{z}_t|\mathbf{x}) \nabla_{\mathbf{z}_t} \log q(\mathbf{z}_t|\mathbf{x}) d\mathbf{x} \right\rangle \, d\mathbf{z}_t \tag{78}$$

$$= \int_{\mathbf{x}} \int_{\mathbf{z}_t} q(\mathbf{x}) q(\mathbf{z}_t|\mathbf{x}) \langle \sqrt{\mathbf{c_2}} \mathbf{s_\theta}(\mathbf{z}_t), \sqrt{\mathbf{c_1}} \nabla_{\mathbf{z}_t} \log q(\mathbf{z}_t|\mathbf{x}) \rangle \, d\mathbf{z}_t d\mathbf{x} \tag{79}$$

$$= \mathbb{E}_{q(\mathbf{x},\mathbf{z}_t)} \left[ \langle \sqrt{\mathbf{c_2}} \mathbf{s_\theta}(\mathbf{z}_t), \sqrt{\mathbf{c_1}} \nabla_{\mathbf{z}_t} \log q(\mathbf{z}_t|\mathbf{x}) \rangle \right] \tag{80}$$

So, only the first term of the right-hand sides of Equation 71 and Equation 72 are not equal. It follows that:

$$\frac{1}{2} \mathbb{E}_{q(\mathbf{z}_t)} \left[ || \sqrt{\mathbf{c_1}} \nabla_{\mathbf{z}_t} \log q(\mathbf{z}_t) - \sqrt{\mathbf{c_2}} \mathbf{s_\theta}(\mathbf{z}_t) ||_2^2 \right] = \frac{1}{2} \mathbb{E}_{q(\mathbf{x},\mathbf{z}_t)} \left[ || \sqrt{\mathbf{c_1}} \nabla_{\mathbf{z}_t} \log q(\mathbf{z}_t|\mathbf{x}) - \sqrt{\mathbf{c_2}} \mathbf{s_\theta}(\mathbf{z}_t) ||_2^2 \right]$$
$$+ \text{ constant} \tag{81}$$

where $\text{constant} = \frac{1}{2} \mathbb{E}_{q(\mathbf{z}_t)} \left[ || \sqrt{\mathbf{c_1}} \nabla_{\mathbf{z}_t} \log q(\mathbf{z}_t) ||_2^2 \right] - \frac{1}{2} \mathbb{E}_{q(\mathbf{x},\mathbf{z}_t)} \left[ || \sqrt{\mathbf{c_1}} \nabla_{\mathbf{z}_t} \log q(\mathbf{z}_t|\mathbf{x}) ||_2^2 \right]$ is constant w.r.t. the energy-based model (EBM) $E()$. In the special case where $\sqrt{\mathbf{c_1}} = \sqrt{\mathbf{c_2}}$, we have:

$$\frac{1}{2} \mathbb{E}_{q(\mathbf{z}_t)} \left[ || \sqrt{\mathbf{c_1}} (\nabla_{\mathbf{z}_t} \log q(\mathbf{z}_t) - \mathbf{s_\theta}(\mathbf{z}_t)) ||_2^2 \right] = \frac{1}{2} \mathbb{E}_{q(\mathbf{x},\mathbf{z}_t)} \left[ || \sqrt{\mathbf{c_1}} (\nabla_{\mathbf{z}_t} \log q(\mathbf{z}_t|\mathbf{x}) - \mathbf{s_\theta}(\mathbf{z}_t)) ||_2^2 \right]$$
$$+ \text{ constant} \tag{82}$$

Therefore, minimizing the first term on the right-hand side of Equation 82 w.r.t. $E()$ (a denoising score matching objective with differently weighted dimensions) is equivalent to minimizing the left-hand side of Equation 82 w.r.t. $E()$. From this equation, it is clear that at the optimum of this DSM objective, for any positive $\mathbf{c}_1$:

$$\mathbf{s_\theta^*}(\mathbf{z}_t) = \nabla_{\mathbf{z}_t} \log q(\mathbf{z}_t) \tag{83}$$

If the score model is parameterized as the gradient of an EBM $E()$, then this implies that for all $t \in [0, 1]$:

$$\exp(-E^*(\mathbf{z}_t; t)) \propto q(\mathbf{z}_t) \tag{84}$$

So, when optimizing for the diffusion loss, the EBM $E(.; t)$ will approximate the correct marginals corresponding the inference model.

## M   Additional samples from our models

We include additional uncurated random samples from our unconditional models trained on CIFAR-10, 32x32 Imagenet, and 64x64 Imagenet. See Figures 9, 10, and 11.

## N   Lossless compression

For a fixed number of evaluation timesteps $T_{eval}$, our diffusion model in discrete time is a hierarchical latent variable model that can be turned into a lossless compression algorithm using bits-back coding [Hinton and Van Camp, 1993]. Assuming a source of auxiliary random bits is available alongside the data, bits-back coding encodes a latent and data together, with the latent sampled from the approximate posterior using the auxiliary random bits. The net coding cost of bits-back coding is

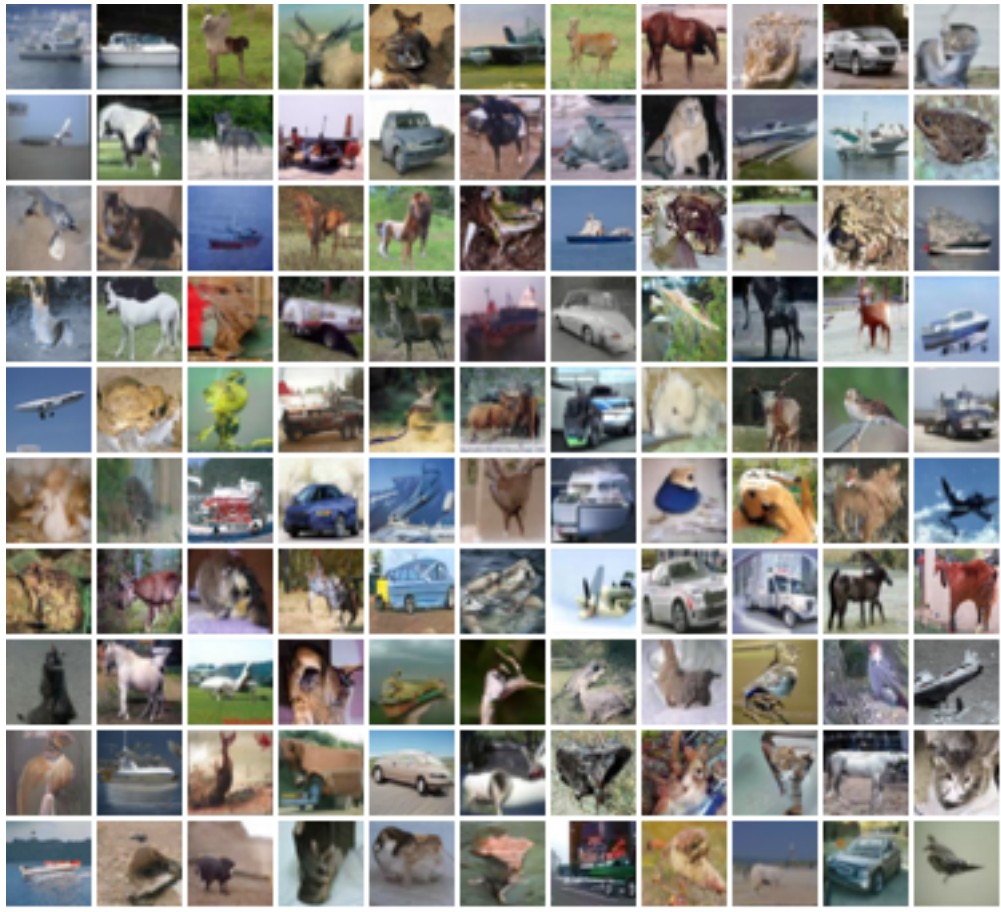

Figure 9: Random samples from an unconditional diffusion model trained on CIFAR-10 for 2 million parameter updates. The model was trained in continuous-time, and sampled using $T = 1000$.

given by subtracting the number of bits needed to sample the latent from the number of bits needed to encode the latent and data using the reverse process, so the negative VLB of our discrete time model is the theoretical expected coding cost for bits-back coding.

As a proof of concept for lossless compression using our model, Table 2 reports net codelengths on the CIFAR10 test set for various settings of $T_{eval}$ using BB-ANS [Townsend et al., 2018], a practical implementation of bits-back coding based on asymmetric numeral systems [Duda, 2009]. Since diffusion models have Markov forward and reverse processes, we use the Bit-Swap implementation of BB-ANS [Kingma et al., 2019]. Practical implementations of bits-back coding must discretize continuous latent variables and their associated continuous probability distributions; for simplicity, our implementation uses a uniform discretization of the continuous latents and their associated Gaussian conditionals from the forward and reverse processes. Additionally, we found it crucial to encrypt the ANS bitstream before each decoding operation to ensure clean bits for sampling from the approximate posterior; we did so by applying the XOR operation to the ANS bitstream with pseudorandom bits from a fixed sequence of seeds. For example, without cleaning the bitstream using encryption, compressing a batch of 100 examples using $T_{eval} = 250$ costs 2.74 bits per byte, but with encryption, the cost improves to 2.68 bits per dimension.

For a small number of timesteps $T_{eval}$, our bits-back implementation attains net codelengths that agree closely with the negative VLB, but there is some discrepancy for large $T_{eval}$. This is due to inaccuracies in the compression algorithm to represent discretized Gaussians with small standard deviations, and small discrepancies in codelength compound into a gap of up to 0.05 bits per dimension when $T$ is large. (In prior work, e.g. [Kingma et al., 2019, Ho et al., 2019b, Townsend et al., 2020], practical implementations of bits-back coding have been tested on latent variable models

with only tens of layers, not hundreds.) In addition, a large number of timesteps makes compression computationally expensive, because a neural network forward pass must be run for each timestep. Closing the codelength gap with an efficient implementation of bits-back coding for a large number of timesteps is an interesting avenue for future work.

## O   Density estimation on additional data sets

At the request of one of the reviewers we also ran our model on additional data sets of higher resolution, less diverse, images. Specifically, we obtain a test set likelihood of 2.14 bits per dim on CelebA-HQ [Karras et al., 2017], and 1.44 on LSUN bedrooms [Yu et al., 2015], both at $128 \times 128$ resolution. Since these are not established benchmarks in density estimation, and since downsampling methods in the literature are not consistent, we don't compare against previous methods for these data sets. Our results are provided purely to give a ballpark estimate of how well our proposed method scales to higher resolution images.

The model used for these data sets is based on that used for Imagenet $64 \times 64$, with an additional level in the UNet at resolution $128 \times 128$, consisting of 16 residual layers using 128 channels. Our model downsamples between the $128 \times 128$ and $64 \times 64$ resolutions, similar to e.g. Ho et al. [2020], but unlike the models we used for the other data sets we considered.

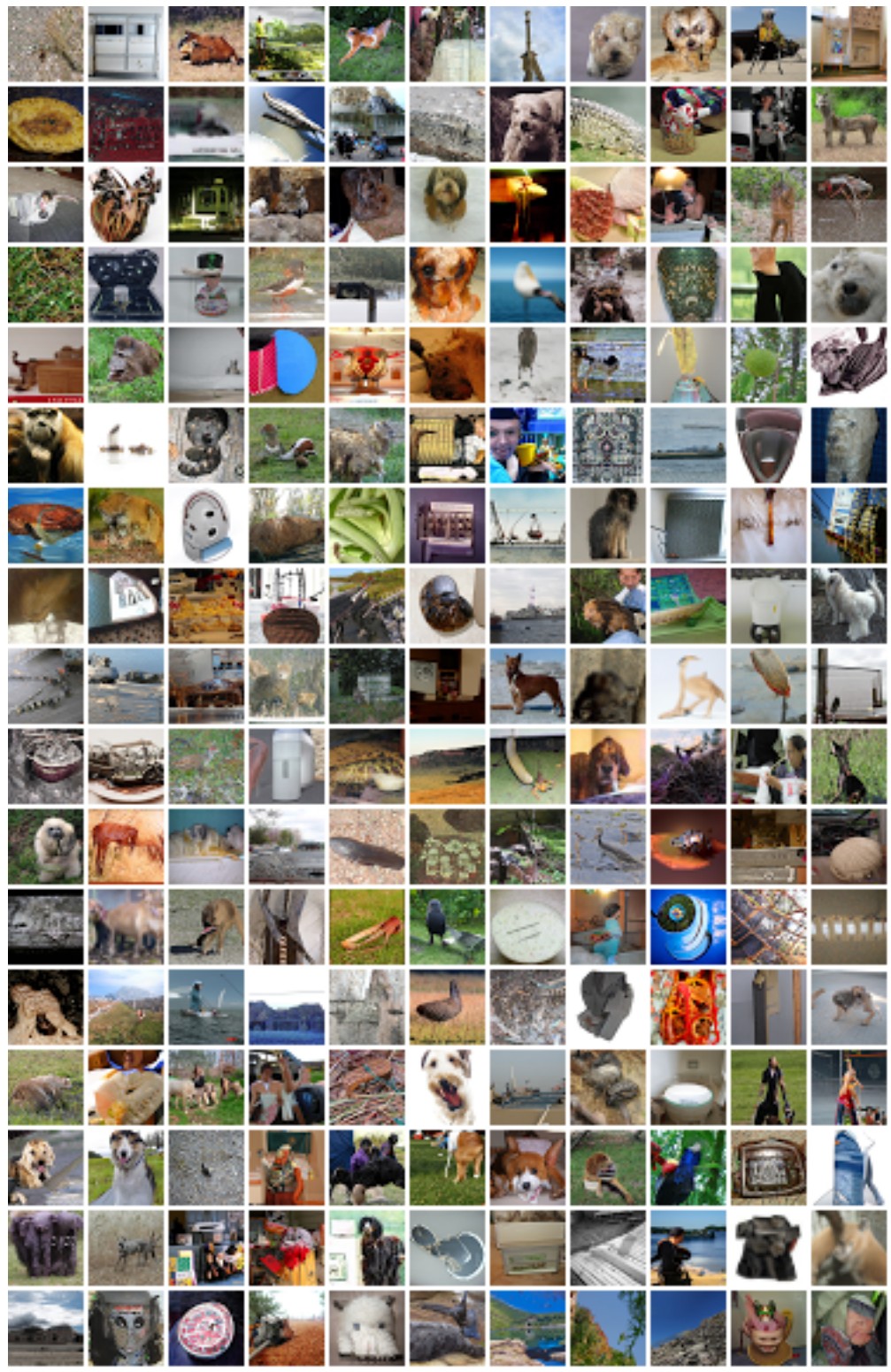

Figure 10: Random samples from an unconditional diffusion model trained on 32x32 ImageNet for 3.7 million parameter updates. The model was trained in continuous-time, and sampled using $T = 1000$.

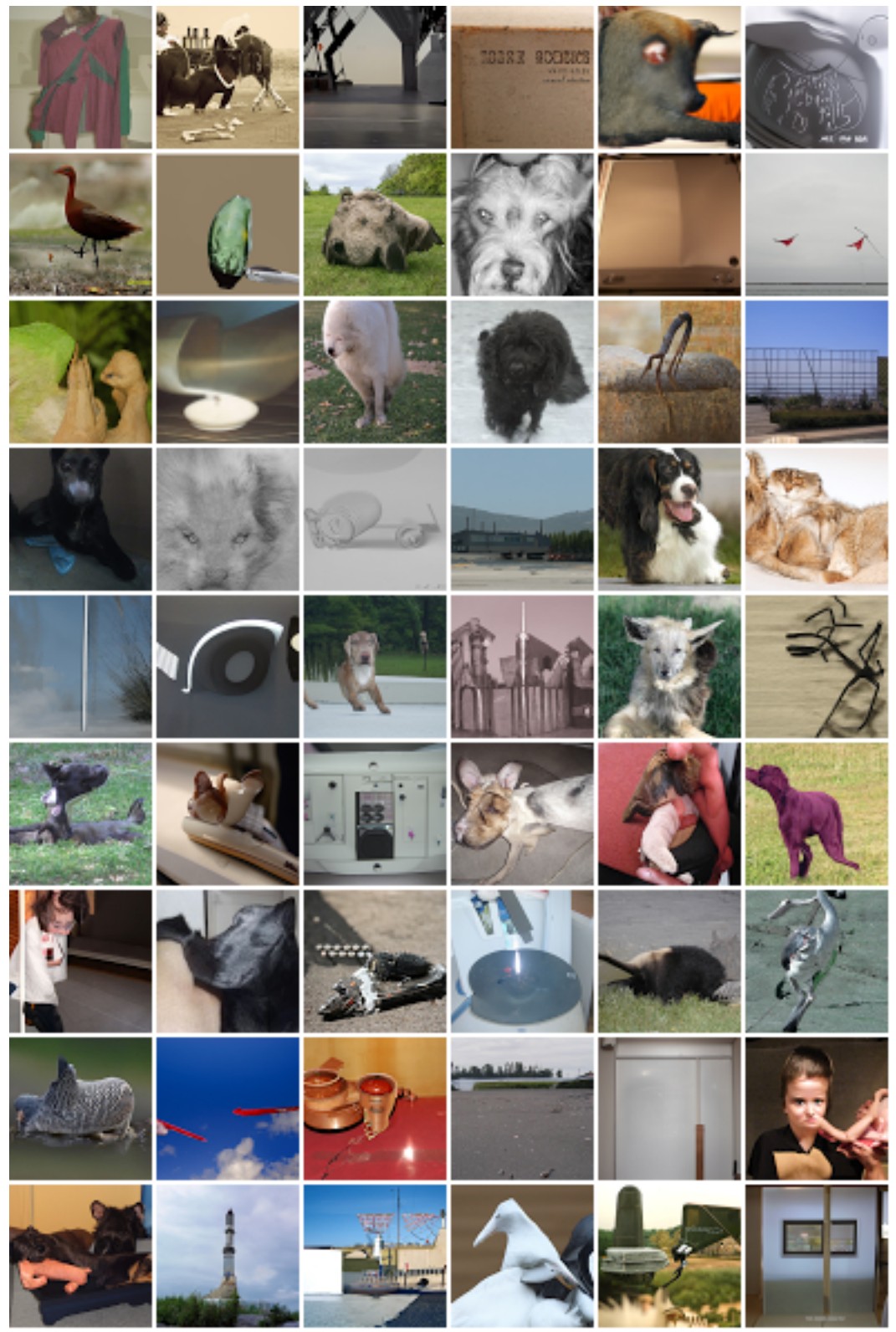

Figure 11: Random samples from an unconditional diffusion model trained on 64x64 ImageNet for 2 million parameter updates. The model was trained in continuous-time, and sampled using $T = 1000$.