# OpenReview forum: "Variational Diffusion Models"
_NeurIPS.cc/2021/Conference — NeurIPS 2021 Poster_

### Official Review · Reviewer_m8xn · 2021-07-04

**Rating:** 6
**Confidence:** 3

**Summary:**

The paper proposed an improvement to DPM. The main contribution is summarized below.

- Sec 3: The diffusion and reverse processes in DPM are re-written with the proposed SNR function. It can be seen as a high-level framework of the noise schedule.
- Sec 4: The ELBO (for both discrete and continuous DPMs) is then re-written in terms of SNR. The authors had a few other findings: e.g., a larger T is better, and equivalence between continuous DPMs.
- Sec 5: The SNR function is parameterized by a neural network and targets at minimizing the estimation variance of ELBO.
- Sec 6: Additional features computed from Fourier transformation are added to data.
- Sec 7: Experiments on CIFAR and ImageNet show improvements on BPD.



**Limitations And Societal Impact:**

Limitations and social impact are discussed in the paper. More questions on limitations are in the main review.

**Main Review:**

The paper is original to my knowledge. The research problem is significant, as DPMs are recently found to be very powerful in a number of generative tasks. As for quality and clarity, generally speaking, the theoretical results are correct, the experimental results are reasonable, but the writing is not very clear (detailed comments below).

Pros:
- The SNR function is the most interesting and important contribution of the paper. It provides a high-level view of both discrete and continuous DPMs. Learning SNR is meaningful as most current noise schedules are heuristic and it is unclear how to setup a best schedule.
- There are some interesting findings (e.g. "more steps is better" and "equivalence between continuous DPMs") and other useful tools (e.g. unbiased estimators and numerically stable computation).
- Experimentally, it is shown that the proposed Fusion, with the SNR learned and additional Fourier features, lead to SOTA BPD.

Cons / main concerns:

My first major concern is that neither the theoretical or the empirical contribution is sufficient. Do the authors consider this paper as a theoretical paper or an empirical paper? Currently, it looks only like a somewhat smart algorithm, with limited theoretical understanding and some experimental validation.
- If the authors want to write a **theoretical** paper, then there should be a **complete** theory for the proposed method and relative findings. However, there is no theorem/proposition in the main text or the appendix. More importantly, many findings are not formally justified, so they are only insights. Below are some suggestions for writing a theoretical paper.
    - Line 104-105: small SNR(1) -> $q(z_1|x)\approx$ the standard Gaussian (need theoretical analysis).
    - Line 109-111: large SNR(0) -> $p(x|z_0)\approx q(x|z_0)$ (need theoretical analysis).
    - Line 152-155 and the paragraph after eq(48): $L_{2T} < L_T$. It makes sense that loss tends to be larger because $z_{t}$ is noisier than $z_{t'}$. However, a formal theoretical justification, even under some more assumptions, is necessary. Then, a tighter (non-zero) bound on $L_T - L_{2T}$ in terms of SNR is ideal.
    - Section 5: The SNR function is learned so that it minimizes the ELBO estimation variance. To make the theory sound, I suggest providing an upper bound on the variance term, which should be relevant to the SNR function or the general noise schedule. Then, provide a theoretical bound on the possible improvement of the variance estimation.
    - Appendix E: a sensitivity analysis for the numerical stable computation is recommended.
- If otherwise, the authors want to write an **empirical** paper, then there should be **extensive** experiments. Below are some suggestions for making it an empirical paper.
    - Evaluation: The BPD improvement is marginal, given that those for prior DPMs are just upper bounds. The FID on CIFAR is worse than prior DPMs. Beating prior methods is not absolutely necessary; however, given these results, it is too early to call the proposed method "SOTA". There should be at least some aspect that your method clearly outperforms other models. In addition, in line 267-268, those DPMs are not *specifically* designed for FID, and if you can get much better FID then you should report the setting and numbers.
    - More datasets should be surveyed, because different datasets have different characteristics. In this paper, both CIFAR and ImageNet are very diverse datasets. How about datasets that are less diverse but have higher quality (resolution), such as CelebA-HQ and LSUN? It would be more interesting to look at other domains, such as audio or 3d point cloud generation.
    - Figure 4 shows that the learned SNR leads to very small variance. However, it is necessary to directly compare BPD between learned SNR and prior (linear/cosine) schedules. What is the improvement of BPD if we just learn the SNR and keep all other settings the same? In addition, why does reducing the variance leads to improvement of BPD, if that is the case?
    - I notice that in Figure 3, having "variance minimization" very marginally improves BPD, whereas having Fourier features significantly improves BPD. How about applying these features to prior DPMs? It seems this is the main reason for the improvement.
    - In Table 2, when $T_{\mathrm{eval}}$ is 100 or 1000, the discrete DPM is better than the continuous DPM. Does that conflict with line 157-158?

Another major concern is about writing. The whole paper is very hard to follow.

- Intro is too short and lacks some key discussion. Why are we interested in image density estimation benchmarks (rather than e.g. FID)? What are the main drawback of current methods? What are the challenges / difficulties, and how do you get around with it? How is the proposed method meaningful for the community? What can people learn from it?
- Always define a symbol **before** using it. (Minor concern for equations: remember to add commas or dots after an equation, such as in eq 8 and 9.)
- Eq(1) seems coming from nowhere. More background is needed. You could start with either diffusion in physics or DDPM by Ho et al 2020. Also, $\mathcal{N}(z_t;\alpha_tx,\sigma_t^2I)$. Why don't you model the covariance matrix in a similar as improved DDPM?
- Line 80-85: The correct order is defining the parameters first and then proving the diffusion process is Markov. Then, why do we care about $q(z_s|z_t,x)$? To make it clear, the ELBO expansion should be introduced here. And similarly, a proof is needed. To make this part clearer, I suggest start with DDPM and show how to adapt the theory here (by talking about connections and modification).
- Line 86-90: It is okay to postpone theory to section 4, but more explanation on the SNR is preferred. Briefly talk about its meaning and how it affects the diffusion process.
- Eq (7): Why do you separate $x$ and $z_0$? In DDPM they are the same thing. What is the benefit here? (Minor concern for notations $s(i)$ and $t(i)$: sometimes the index is ignored, e.g. in eq17 and eq47. I suggest use subscriptions, and also remember to apply it to $t'$ as in eq47.)
- Line 106-111: this part is confusing. Where did you define $p(x|z_0)$? Why can you factorize it according to dimensions? Why does it approximate the $q(x|z_0)$?
- Line 112-116: What is the network and what are the re-parameterizations? Why does $\hat{x}_{\theta}$ predict $x$? And similarly, it is preferred to connect eq11 and eq12 to DDPM as well as eq4.
- Eq 13: what is $W_t$?
- A general problem for section 3 is that too many things are postponed to later sections without any further explanation. This makes it very hard to read for general readers.
- Section 4.2: Do $L_T$ and $L_{2T}$ use the same SNR, or you are taking the inf over all SNR? And as mentioned before, line 151-154 is not rigorous.
- Line 221-222: What random variables are the $\mathbb{E}$ and VAR over? $t$, $\epsilon$, or both? (Minor concern: in eq 50, it should be $\frac{\partial}{\partial\eta}$.)
- Section 5: As mentioned before, why does reducing variance help? A deeper discussion is needed.


-----------------------------------------

In the rebuttal, the authors added insights and preliminary experimental results (or descriptions) to support their methods. These sound reasonable to me so I will raise my score to 6.


**Time Spent Reviewing:**

10 hours

---

> ### Author Response · Authors · 2021-08-10
> **Author Response**
>
> Thank you for your thorough review. We appreciate your specific suggestions for improvement and detailed criticisms, and we will incorporate them in the final version of the paper.
>
> A major objection is that the paper presents both theoretical and empirical contributions, while not formulating explicit theorems or providing a full set of formal proofs. We believe our paper presents several theoretical insights into the design and training of diffusion models. While not a complete theory, these results have guided our experiments and led to promising empirical performance on density modeling for images. We agree that all statements made should be justified, and we believe they are in our paper.
>
> Regarding your specific concerns:
>
> - About needing more theoretical analysis for Lines 104-105 and 109-111 about small SNR(1) and large SNR(0).
>
> We will add a more complete analysis in the appendix. We did mention the intuition for the large SNR(0) case (the influence of the unknown data distribution q(x) is overwhelmed by the likelihood q(x|z_0)), and the small SNR(1) case is well discussed in previous papers on DDPM.
>
> - Line 152-155 and the paragraph after eq(48): L_2T < L_T. It makes sense that loss tends to be larger because z_t is noise then z_t’. However, a formal theoretical justification, even under some more assumptions, is necessary. Then, a tighter (non-zero) bound on L_t - L_2T in terms of SNR is ideal.
>
> Here we state that more steps is better, i.e. formally L_2T < L_T, if the original data x is better predicted by our denoising model from the less noisy data z_t’ then from the noisier data z_t. This could be quantified under very specific assumptions on the data distribution and model, and a bound could be provided, but we do not agree that this would provide any additional insight into the behavior of diffusion models in the real world. We thus prefer to keep the original claim and formulation unchanged, which captures the core insight and which is formally supported by our analysis.
>
> - Section 5: The SNR function is learned so that it minimizes the ELBO estimation variance. To make the theory sound, I suggest providing an upper bound on the variance term, which should be relevant to the SNR function or the general noise schedule. Then, provide a theoretical bound on the possible improvement of the variance estimation.
>
> We believe that providing meaningful and useful bounds here would require making very restrictive assumptions about the data distribution and model. We believe our empirical evaluation in Table 2b provides more insight into the variance reduction that can be achieved in practice.
>
> - Evaluation: The BPD improvement is marginal, given that those for prior DPMs are just upper bounds. The FID on CIFAR is worse than prior DPMs. Beating prior methods is not absolutely necessary; however, given these results, it is too early to call the proposed method "SOTA". There should be at least some aspect that your method clearly outperforms other models. In addition, in line 267-268, those DPMs are not specifically designed for FID, and if you can get much better FID then you should report the setting and numbers.
>
> Our numbers for the likelihood are also just bounds, similar to those for prior DPMs. Even though our numbers understate the quality of our models, they nonetheless beat prior methods, including other techniques with fully tractable likelihoods, by a significant margin. We thus feel justified in claiming SOTA on these density estimation benchmarks. Our models are indeed not currently SOTA in terms of FID: We are already very explicit about this in the paper, but will make this even more clear in the final version of the paper.
>
> - More datasets should be surveyed, because different datasets have different characteristics. In this paper, both CIFAR and ImageNet are very diverse datasets. How about datasets that are less diverse but have higher quality (resolution), such as CelebA-HQ and LSUN? It would be more interesting to look at other domains, such as audio or 3d point cloud generation.
>
> In the literature so far CelebA-HQ and LSUN have not been popular benchmarks for density estimation, and hence there are no strong baselines for us to compare against on these datasets. We do agree that setting new baselines on these data sets would be a very useful contribution to the literature, and we will add results for these data sets to the appendix for the final version of the paper  (these models are currently training but unlikely to be finished by the end of the rebuttal period).
>
> - Figure 4 shows that the learned SNR leads to very small variance. However, it is necessary to directly compare BPD between learned SNR and prior (linear/cosine) schedules. What is the improvement of BPD if we just learn the SNR and keep all other settings the same? In addition, why does reducing the variance leads to improvement of BPD, if that is the case?
>
> We provide exactly this requested comparison in Table 2b.
> We expect reducing the variance to speed up training of the model, but not necessarily to change the end result in terms of BPD. If space allows, we will add more discussion about this in the final version.
>
> - I notice that in Figure 3, having "variance minimization" very marginally improves BPD, whereas having Fourier features significantly improves BPD. How about applying these features to prior DPMs? It seems this is the main reason for the improvement.
>
> Prior DPMs do not benefit much from Fourier features since they do not model the data at very high SNR, which is necessary for obtaining the best likelihoods and which is where these features are most useful. By learning the noise schedule, our model naturally picks the SNR_max that’s optimal to use with or without Fourier features. We will add additional experiments that show this in the final version.
>
> - In Table 2, when T_eval  is 100 or 1000, the discrete DPM is better than the continuous DPM. Does that conflict with line 157-158?
>
> No, we show that more steps (or infinite steps) is better when keeping the SNR function fixed. The numbers in Table 2 for low T_eval have SNR schedules that are specifically optimized for these settings, while the continuous model has not been optimized for this setting.
>
> - Intro is too short and lacks some key discussion. Why are we interested in image density estimation benchmarks (rather than e.g. FID)? What are the main drawback of current methods? What are the challenges / difficulties, and how do you get around with it? How is the proposed method meaningful for the community? What can people learn from it?
>
> We had a hard time fitting all of our most important content within the 9 page limit. We agree that a more thorough discussion would be useful here, and will try to incorporate it in the final version if space allows.
>
> - Equation 1 [..] Why don't you model the covariance matrix in a similar as improved DDPM?
>
> This equation describes the forward diffusion process, where improved DDPM only models (the diagonal of) covariance in the reverse diffusion model. As they describe in their paper, this is useful only when the number of steps is small. We instead choose to focus on the continuous time case where modeling the reverse variance would thus not add anything.
>
> - Line 106-111: this part is confusing. Where did you define p(x|z_0)? Why can you factorize it according to dimensions? Why does it approximate the q(x|z_0)?
>
> p(x|z_0) is a product between the unknown data density / prior p(x) and the factorized likelihood q(z_0|x). For high SNR(0) the likelihood overwhelms the prior, and hence the posterior p(x|z_0) becomes tractable and factorizable, and equal to our choice of q(x|z_0). We will add additional discussion about this to the paper.
>
> - Line 112-116: What is the network and what are the re-parameterizations? Why does \hat{x} predict x? And similarly, it is preferred to connect eq11 and eq12 to DDPM as well as eq4.
>
> The network and parameterization is discussed in Appendix A.1. We will add a pointer to this discussion in the main text.
>
>  - Section 4.2: Do L_T and L_2T use the same SNR, or you are taking the inf over all SNR?
>
> Yes, same SNR. We will make sure this is explicit.
>
> - Line 221-222: What random variables are the \E and VAR over? t, eps, or both?
>
> Both. We have made this more explicit.
>
> - Section 5: As mentioned before, why does reducing variance help? A deeper discussion is needed.
>
> We will add this discussion, so far as the page limit will allow it.
>
> - Various writing / notation tips.
>
> Thanks, these are very useful. We’ll do our best to incorporate everything into the final version of the paper.

---

> > ### Comment · Reviewer_m8xn · 2021-08-14
> > **Reviewer response**
> >
> > I thank the authors for their comprehensive feedback. Much of the feedback addressed my concerns. I also appreciate that the authors plan to conduct more experiments to support some of their key findings.
> >
> > I disagree with authors' response to the theoretical concerns. Good theory should be able to explain and predict what helps and gives us the correct insight. Nevertheless, as the authors plan to provide more experimental validation to support their key findings, my theoretical concerns will not affect my final decision.
> >
> > Regarding the experiments, I have several follow-up questions for the authors.
> >
> > - Regarding variance minimization, the authors respond "We provide exactly this requested comparison in Table 2b. We expect reducing the variance to speed up training of the model, but not necessarily to change the end result in terms of BPD." Do you mean Table b in Fig 4? What is the insight behind the claim that "reducing the variance can speed up training but will not change the final BPD"? Will you conduct experiments to support this claim?
> >
> > - Regarding "more steps is better", the authors respond "The numbers in Table 2 for low T_eval have SNR schedules that are specifically optimized for these settings, while the continuous model has not been optimized for this setting." I think the authors need to clarify what does "more steps is better" mean for the continuous model. Are you claiming that the continuous model evaluated on (1) T_eval steps, or (2) infinite steps is better than a discrete model trained and evaluated on T_eval steps? If it's (1), then I think Table 2 is not conclusive. Is it possible to conduct an experiment to show this? If it's (2), then Table 2 already provides the answer. However, does it limit the benefit of using the continuous model?
> >
> > - Regarding the Fourier transformation, the authors respond "Prior DPMs do not benefit much from Fourier features since they do not model the data at very high SNR, which is necessary for obtaining the best likelihoods and which is where these features are most useful. By learning the noise schedule, our model naturally picks the SNR_max that’s optimal to use with or without Fourier features. We will add additional experiments that show this in the final version." The insight on why prior models do not benefit from Fourier features makes sense to me. Can you briefly describe how you will prove that the model picks the optimal SNR_max?
> >
> > - Regarding line 269-270. I still hope the authors can report the numbers if you claim that it's possible to get significant better FIDs. This will give readers a sense on how well the model can perform when evaluated with FID. Otherwise I don't think it's appropriate to put this sentence in the main text.
> >
> > My score can be adjusted based on authors' response.

---

> > > ### Author Response · Authors · 2021-08-18
> > > **author response**
> > >
> > > Thank you for the follow up.
> > >
> > > * _Regarding variance minimization:_ Sorry, yes our answer was unclear. Table 4b shows the effect of learning the noise schedule on the variance of the VLB estimator, as compared to the DDPM and IDDPM schedules from the literature. For this experiment the model parameters and SNR endpoints were held fixed, so the BPD is identical. Regarding your question about how learning the SNR during training impacts final achieved BPD, we found the following:
> > > 1. When using the original DDPM model, without Fourier features, learning the SNR still reduces variance but does not help significantly improve BPD. Without Fourier features the model is not able to learn the fine details of the data. SNR_max therefore stays low during training, and BPD stays high. We'll include this experiment in the next update (most likely in the appendix).
> > > 2. When using our model in combination with the original DDPM schedule, the inclusion of Fourier features does not improve the BPD much, since the SNR_max is still fixed to a low value. We'll also include this experiment.
> > > 3. In Figure 3, the model with "no variance minimization" does not learn the shape of the SNR schedule, but it does learn the endpoints SNR_min and SNR_max (and uses log-linear interpolation inbetween). As shown in Figure 3, for a given number of optimization steps the resulting BPD is higher in this case than for the model with variance minimization enabled, but at the far right we can see the lines for both setting converge towards the same final BPD. Anecdotally, we find that the convergence in final BPD continues if we run the model for longer than shown in the figure. Our interpretation of this is that having lower variance loss estimates speeds up optimization, but it does not change the capacity of the model, and hence does not change the final BPD than can be reached.
> > >
> > > * _Regarding "more steps is better":_ What we show analytically is that more steps is better for a discrete-time model, where we keep the SNR schedule fixed as we add more steps. (so the additional steps interpolate the old SNR values) In the next update we'll include a slightly cleaner proof for this, as well as a new figure that explains this graphically. We can thus also say that the continuous-time model evaluated on an infinite number of steps will be better than the discrete-time model trained and evaluated on a smaller discrete number of steps T_eval. (so statement (2) in your question above). For small finite T_eval, we would generally expect the discrete-time model to be better than the continuous-time model, if T_train = T_eval, since it then explicitly optimizes for this evaluation setting (so statement (1) in your question is not something we claim). As to whether this limits the benefit of the continuous-time model: As Table 2 shows, the discrete and continuous models indeed behave very similarly if T_eval is large. However, in this case the continuous-time model stil gives us  1: a way to minimize the variance of the loss estimate by learning the SNR schedule, which allows us to train faster,  2: a simplified implementation of the training loss,  3: flexibility when evaluating with a different number of steps than used during training, i.e. T_eval != T_train.
> > >
> > > * _[...] Can you briefly describe how you will prove that the model picks the optimal SNR_max?_ In our case SNR_min and SNR_max are free parameters that we learn by optimizing the ELBO. (This is distinct from learning the schedule inbetween the endpoints, which does not optimize the ELBO but instead minimizes variance) As long as SGD is able to solve the optimization problem (which is not strictly guaranteed of course), the model will thus pick the optimal SNR_max. Figure 3 already includes runs where we learn the SNR_min and SNR_max in this way, with and without Fourier features: To this we will add details on the SNR endpoints that end up being learned, and we'll add a comparison to the SNR_min and SNR_max corresponding to prior DPMs.
> > >
> > > * _Regarding line 269-270 about FID_ We are currently still running experiments to see how far we can improve FID. What we found so far is that we can improve the FID on CIFAR-10 to about 4.0 by using the DDPM SNR weights (see Appendix F) in combination with a U-net architecture that's closer to the original DDPM. We hope to fully close the gap to the original DDPM by the time the camera ready paper is due. In any case, we'll update the text to reflect what we find.

---

> > > > ### Comment · Reviewer_m8xn · 2021-08-21
> > > > **Reviewer response**
> > > >
> > > > Thank the authors for the response.
> > > >
> > > > The insights and the preliminary experimental results (or descriptions) sound very reasonable to me. I intend to increase my score to 6.

---

### Official Review · Reviewer_bEoD · 2021-07-07

**Rating:** 6
**Confidence:** 5

**Summary:**

The paper introduce a new insight to diffusion generative models. Mainly, the paper re-write the forward and backward processes in new formulation which incorporate the SNR value into the diffusion process.

Previous diffusion process like DDPM can be regard as instance of the proposed model. The snr value in the proposed method is learned jointly with the model whereas previous approach use fix snr value.

The paper showed various theoretical results: (i) More steps is better, (ii) for continues time loss the model is invariance to the noise schedule, (iii) the training is impacted the start and end snr values.
The paper introduce to main novelties: (i) learning noise schedule and (ii) insert Fourier features as additional features


**Limitations And Societal Impact:**

The authors addressed the limitations and the potential negative social impact of their work.

**Main Review:**

The paper is well written as show big progress in understanding diffusion process. However i concern with the following issues:

(1) The paper introduce results only for low-mid resolution datasets (64x64<). Does the proposed model improved the results for larger resolution datasets, like LSUN 256x256?

(2) The results in the paper is given with bits per dim. Since most of the literature in diffusion process use FID score, it would be beneficial if the paper also show the improvement in terms of FID scores.

(3) The paper should add ablation analysis to see the contribution of the learned noise schedule.

(4) In the ablation study, it seems that adding the Fourier features give the major part of the bits per dim improvement. Therefore, it will be beneficial to check what is the performance of other diffusion method when adding the Fourier features, e.g. Improved DDPM with Fourier features

(5) There are some typo in Eq22 (2nd line), v-->v' at the beginning of the line.


**Time Spent Reviewing:**

5 hours

---

> ### Author Response · Authors · 2021-08-10
> **Author Response**
>
> We thank the reviewer for his/her time and effort reviewing our submission.
>
> Responses to numbered questions:
>
> **Reviewer: "The paper introduce results only for low-mid resolution datasets (64x64<). Does the proposed model improved the results for larger resolution datasets, like LSUN 256x256?"**
>
> We have only performed experiments on the CIFAR-10, ImageNet 32x32 and ImageNet 64x64 dataset. We see no reason to doubt that the proposed method would also work well for less diverse but more high-resolution datasets, such as LSUN 256x256, but have not done experiments with such datasets.
>
> **Reviewer: "The results in the paper is given with bits per dim. Since most of the literature in diffusion process use FID score, it would be beneficial if the paper also show the improvement in terms of FID scores."**
>
> We focussed our paper on likelihood, but we will be more explicit about the tradeoff between likelihood and FID scores.
>
> **Reviewer: "The paper should add ablation analysis to see the contribution of the learned noise schedule."**
>
> We just ran an ablation experiment, and using the fixed noise schedule from DDPM instead of our learned noise schedule, we see results that are significantly worse; more than 1 BPD for CIFAR-10. In the updated version of our draft, we will include our ablated numbers.
>
> In Table 2b we have an ablation study showing that our learned noise schedule in continuous time leads to large reductions in the variance of the ELBO estimator. As discussed, changing the noise schedule (between end points) in this setting does not change the objective, so we don’t see a significant difference in the final result if trained for a long enough time.
> In discrete time the learnable noise schedule does have a positive impact on the final BPD achieved, but this is hard to disentangle from our other contributions (Fourier features, our choice of variance in p(), parameterization in terms of SNR instead of t, etc) since these were all necessary to be able to successfully learn the noise schedule in a stable way.
>
> **Reviewer: "In the ablation study, it seems that adding the Fourier features give the major part of the bits per dim improvement. Therefore, it will be beneficial to check what is the performance of other diffusion method when adding the Fourier features, e.g. Improved DDPM with Fourier features"**
>
> We find that the main benefit of Fourier features is that it allows us to successfully model the data at much higher signal to noise ratios than those considered by previous methods. Applying Fourier features to improved DDPM without also changing the noise schedule (which does not place heavy weight on high SNR) does not yield large improvements in bits per dim, as we will show in our final version of the paper. Adding Fourier features also does not improve the FID score of this model.
>
> **Reviewer: "There are some typo in Eq22 (2nd line), v-->v' at the beginning of the line."**
>
> Thanks. This equation (bottom left) is indeed missing two ‘, which we’ll correct in the next version.

---

### Official Review · Reviewer_Mjaj · 2021-07-12

**Rating:** 7
**Confidence:** 4

**Summary:**

The paper tackles density estimation with denoising diffusion-based generative models and achieves very strong, state-of-the-art likelihoods (bounds) on standard image modeling benchmarks. The formulation largely relies on previous denoising diffusion probabilistic models (DDPMs), but uses the exact ELBO objective without dropping weighting coefficients in the mixture of losses, as was done in DDPM. The paper also proposes to learn the noise schedule, either using the ELBO objective or a variance reduction objective, and adds additional Fourier features into the model, which brings a significant benefit. On the theoretical side, the paper demonstrates how a change of variables in the time variable can lead to an equivalence between models trained with different noise schedules and shows how the important aspect of the noise schedule is only its behaviour at its endpoints.

**Limitations And Societal Impact:**

__(Negative) Societal Impact:__ Has been discussed very briefly, but in a satisfactory manner.

__Limitations:__ I think limitations could have been discussed more thoroughly. The proposed *Fusion* model is a typical denoising diffusion model, and as such has well-known and important limitations, I think. For example:
1. Very slow, iterative synthesis. The model samples shown in the appendix use 1000 synthesis steps, for example.
2. Other types of likelihood-based generative models, such as VAEs and autoregressive models, can be applied on arbitrary data types (language, graphs, discrete/binary data, etc.). Most current diffusion models, including the one in this paper, are limited to continuous data types where we can easily define meaningful diffusion processes.
3. While the quantitative results of the paper are without doubt very strong, there seems to be a trade-off between achieving strong FID scores (or perceptual image quality more generally) and NLL. This can be considered a limitation.

I am aware that this paper does not aim to address these aspects, but I think it would not hurt to acknowledge these important limitations.

**Main Review:**

__Overall Impression:__ The *Fusion* model is mostly based on DDPM [1], while introducing additional tricks and architecture modifications (learning the noise schedule, adding Fourier features) that benefit the quantitative performance. The paper primarily stands out by its extraordinarily strong quantitative results, although the computational resources required for reproducing the results seem intense.

__Originality:__ As mentioned, the core model formulation (section 3) is largely based on DDPM [1], while reformulating certain terms and defining the noise schedule in terms of a signal-to-noise-ratio SNR(t), but it is essentially the same model, I think. The fact that more steps is better, leading to a tighter bound (section 4.2), is well-known and has been pointed out already by [2], which proposed this class of models. The originality and conceptual novelty up to this point seems limited.

The demonstration that in the continuous time formulation the only thing that truly matters about the noise schedule are its endpoints is quite interesting, in particular considering that multiple previous and concurrent works explore different noise schedules. The idea to learn the noise schedule in itself is not novel, as it has been pointed out before [1], but this paper seems to be the first one to actually do it, proposing meaningful objectives for that. Overall, it feels the originality and theoretical/methodological novelty is not impressive, but there are some interesting contributions.

__Clarity:__ The paper is well written and easy to read.

__Specific Questions and Feedback:__
1. In Section 5, in the discretized setting, the noise schedule is learned using the ELBO objective. In that case, I think the model essentially behaves as a VAE. However, VAEs suffer from the prior-hole problem, explaining their poor sample quality. One major advantage of diffusion models is that they usually avoid this problem, diffusing each individual data point to full noise by construction. But by learning the noise schedule with the ELBO, don't we re-introduce this issue? What if the noise schedule learns to add virtually no noise to the data, so the diffusion objective (which is sort of a bunch of reconstruction objectives, given the noisy data) can be easily minimized, while still encoding data points under the prior, as in a VAE, to also satisfy the prior term? I admit that if we are purely targeting strong likelihood this argument doesn't matter. Nevertheless, it seems we may lose one of the main advantages of diffusion models here and sample quality may suffer. I feel this aspect is discussed insufficiently.
2. The authors also propose to instead learn the noise schedule with a variance reduction objective. This makes a lot of sense, but I think this is conceptually analogous to importance sampling techniques proposed by previous work: In principle, we can either optimize the noise schedule as proposed here considering uniform sampling from $t$, or alternatively we can use any arbitrary fixed noise schedule (fixing only the endpoints) and instead use importance sampling that oversamples the important $t$, similarly reducing variance. This seems basically equivalent, explained by the change-of-variables discussed in the paper. But this conceptual relation of learning the noise schedule to previously proposed importance sampling schemes is not discussed in the paper at all.
3. Regarding the theoretical equivalence between different models with different noise schedules: Does this only hold for noise schedules corresponding to a similar underlying SDE? This is, does this mean, for example, that the $\beta(t)$ in the VPSDE [3] is basically arbitrary? Or does this mean that also the underlying form of the SDE itself does not matter (apart from endpoint behavior and for training objective variance)? For example [3] proposed the SubVPSDE. Would it still be helpful to explore other SDEs, considering the continuous time framework, in light of the results?
4. The strongest results are obtained using the continuous setting for both training and evaluation (see Tab. 2). How is evaluation done in this case? Via black-box ODE solvers, leveraging the connection of continuous diffusion models to SDEs and ODEs and therefore continuous Normalizing flows [3]? I feel like I am missing some details regarding the model evaluation. I also didn't find this in the appendix.

__Conclusion:__ I think the paper meets the bar for publication, considering the importance of likelihood-based generative modeling and the paper's very strong quantitative results on that. It also has some minor theoretical contributions that help the understanding of diffusion models and the proposed architectural innovations may be adopted by the community.

[1] Ho et al., "Denoising Diffusion Probabilistic Models";
[2] Sohl-Dickstein et al., "Deep Unsupervised Learning using Nonequilibrium Thermodynamics";
[3] Song et al. "Score-Based Generative Modeling through Stochastic Differential Equations"

**Time Spent Reviewing:**

5

---

> ### Author Response · Authors · 2021-08-10
> **Author Response**
>
> We thank the reviewer for his time and effort reviewing our paper; we highly appreciate it.
>
> The review contains some significant inaccuracies and misrepresents a couple of key aspects of the paper, which we will now address.
>
> **Responses to “Overall Impression” and “Originality”**
>
>  - The reviewer states that the model is “essentially the same model” as DDPM. While the learned generative model has a similar form to DDPM, our models allow for a learnable noise schedule which has not been achieved previously and empirically has proven to be one of the most sensitive design decisions to impact performance in prior work on diffusion models . We also condition the network on SNR(t) instead of t, use Fourier Features as additional inputs, and specify the model in continuous time. As we show experimentally, these changes have a significant impact on the capacity of the model to estimate the data density.
>
>  - The reviewer states: “The idea to learn the noise schedule in itself is not novel, as it has been pointed out before [1], but this paper seems to be the first one to actually do it, proposing meaningful objectives for that.”
> Our solution not only required a different objective, but also a novel parameterization. Earlier works specified the variance of q(z_t|x) through summation or integration of incremental variances of q(z_{t+1}|z_t), which led to unsuccessful attempts at learning the schedule. Our learnable smooth monotonic function that specifies the variance of q(z_t|x) directly, was a key component of the solution.
>
> - The reviewer states: “The fact that more steps is better, leading to a tighter bound (section 4.2), is well-known and has been pointed out already by [2].”
> We reviewed [2] and it does not show that more steps leads to a tighter variational lower bound. We are also not aware of any other works that rigorously show this.
>
> **Responses to “Specific Questions and Feedback”**
>
> 1. The reviewer worries about the “prior hole problem” and asks “What if the noise schedule learns to add virtually no noise to the data”. Adding virtually no noise to the data leads to an extremely high prior loss KL(q(z_0|x)||p(z_0)), so the network avoids such solutions. In practice, the model learned a z_1 that is 99.5% noise, making q(z_1|x) practically indistinguishable from p(z_1). The reported FID is also the best among likelihood-based models. Qualitatively, the samples from our model provided in the main paper and appendix are higher perceptual quality than those from earlier likelihood-based models, including autoregressive models. The reviewer correctly points out that the log-likelihood objective can be sub-optimal for perceptual quality. This is indeed true, and we will do a better job communicating this trade-off in the final draft.
>
> 2. The importance sampling approach cannot be directly applied to the continuous-time objective, and is heuristically designed to reduce variance. In contrast, we directly minimize variance of the estimator and can apply our approach to both discrete and continuous-time diffusion objectives.
>
> 3.  The invariance/equivalence results in Sections 4.3.1 and 4.3.2 indeed show that the choice of SDE, e.g. VPSDE vs SubVPSDE, doesn’t matter for likelihood if the right specification of the model is used. However, we find that if the score network is badly parameterized, it can still matter in practice. For example, [3] conditions the score network on timestep ‘t’, whereas we show that it makes more sense to condition on log(SNR(t)), since otherwise the network needs to unnecessarily infer the SNR from t. Similarly, the network needs to be able to correctly scale its input in order to be invariant to the noise schedule, as shown in Section 4.3.2.
>
> 4. The continuous-time model is simply evaluated using the unbiased ELBO estimator from Equation 24. We do not use black-box ODE or SDE solvers. Hence, the reported bits-per-dim (BPD) numbers are upper bounds. The real BPD is even better, but we do not know by how much. We have tried using the likelihood estimator based on the probability flow ODE from [3], but found it to result in unreliable estimates for our model due to the small amount of noise added by the encoder.
>
> **Response to Limitations**
>
> Thanks for the feedback; we agree that briefly mentioning these limitations would indeed be a useful addition; we’ll try to incorporate within the page limit.

---

> > ### Comment · Reviewer_Mjaj · 2021-08-13
> > **Reply to Author Response**
> >
> > I would like to thank the authors for their comprehensive reply to my review. I thought that it might be helpful to clarify two points that I made in my review and go into a bit more detail.
> >
> > **1.** I made the comment that [2] had already pointed out that more diffusion steps result in a tighter bound. What I was specifically referring to is that [2] point out that "if the forward and reverse trajectories are identical, corresponding to a quasi-static process, then the inequality in Equation 13 becomes an equality" (below Eq. 14 in Section 2.4). However, as had been discussed before in the paper (Sections 2.2 and 2.3 and references), when we have more steps and consequently smaller $\beta_t$'s, we do approach a quasi-static process with similar forward and reverse trajectories. Consequently, one can directly conclude that "more steps are better" in that the bound becomes tighter. That is how I have been interpreting this work. I hope that this helps to understand what I was referring to in my review.
> >
> > **2.** I would also like to follow up regarding the prior-hole question. I very much agree that having almost no noise in the process would be heavily penalized by the prior KL loss and I probably should have worded my question better. But what would happen if the model was trained with much less diffusion steps? I could imagine that in that case the model indeed behaves a bit more like a regular VAE and decides to learn a noise schedule such that the model also "encodes" significant information into the latent variables at $z_1$, in which case $KL(q(z_1|x)||p(z_1))$ might be non-negligibly $>0$ (with $q(z_1|x)$ being narrower than $p(z_1)$), just like the KL term in vanilla VAEs (after all, a hypothetical single-step diffusion model would basically be a regular VAE in which case this KL would almost certainly be $>0$). In principle, one could train different models with different numbers of diffusion steps and check this KL value. Intuitively, I would expect larger KLs with this prior the less steps we have (which could then result in "prior-holes"). Another interesting experiment could be to investigate the value of the $KL(q(z_1|x)||p(z_1))$ together with all the other KL terms that are part of the diffusion (Eq. 16). When we learn the noise schedule, where do we pay most KL penality along the process? I could imagine, for example, that the optimal learnt noise schedule will result in a situation where all KLs are approximately equal, although I am speculating here. I do not necessarily expect you to run these experiments at this point during the reviewing process, but I wanted to clarify these thoughts and think that it could be interesting to run such experiments and add to the paper if the results are insightful. I also want to emphasize that I acknowledge and appreciate the strong empirical evidence that this appears to not be a problem at all for the kind of models with many steps that were trained in the paper towards state-of-the-art likelihoods.
> >
> > I hope this helps!
> >
> > [2] Sohl-Dickstein et al., "Deep Unsupervised Learning using Nonequilibrium Thermodynamics"

---

> > > ### Author Response · Authors · 2021-08-13
> > > **thanks for the clarification**
> > >
> > > Thank you for your clarification.
> > >
> > > 1)
> > > I re-read [2] and checked with its author to make sure I fully understand the claims it makes. What [2] claims is that the bound becomes tight for increasing number of steps, **if the reverse model $p_{\theta}$ is perfect**, i.e. if we have a perfect denoising model, $\hat{x}_{\theta}(z_t) = \mathbb{E}_q(x | z_t)$ in our notation. Although we certainly strive for this, perfection is never achieved in practice. Concurrent to our work, [3] analyses the continuous time VLB and finds that in practice the VLB can actually still be quite loose, even for very good fully-trained continuous-time diffusion models.
> > >
> > > My understanding of the literature is that we did not yet have an answer to the question of whether more or fewer steps is better in practical situations with good, but not perfect, model $p_{\theta}$: Since the continuous time VLB is a limit with an infinite number of KL divergence terms it matters a great deal in practice whether these individual KL divergences go to zero fast enough to actually make the resulting bound improve as T grows large. What we show is that more steps is better under a weaker assumption, namely that on average $\lVert \hat{x}_{\theta}(z_s) - x \rVert_2^2 < $
> > >
> > > $\lVert \hat{x}_{\theta}(z_t) - x \rVert_2^2$ for $s < t$. (sorry, could not get this to render correctly on a single line for some reason)
> > >
> > > Thank you for pointing all of this out: We’ll update to mention the relationship with the claims in [2], and to more clearly describe the subtleties mentioned above.
> > >
> > >
> > > 2)
> > > I think you’re right that the “prior hole problem” can occur when training the model with a small number of steps T. I don’t currently have results on how the learned minimum SNR varies with varying T, but my intuition here matches yours: If the KL divergences at each timestep are approximately equal, then there might also be larger prior holes if T gets small. In any case we do see a deterioration in sample quality as T gets small. Results on $KL(q(z_1|x) | p(z_1))$ for varying T might be obtainable from our previously run experiments: we’ll look into this.
> > >
> > >
> > > [3] Yang Song, Conor Durkan, Iain Murray, and Stefano Ermon. Maximum likelihood training of score-based diffusion models. arXiv e-prints, 2021.

---

> > > > ### Comment · Reviewer_Mjaj · 2021-08-26
> > > > **Thanks - raised my score.**
> > > >
> > > > Thank you very much for the clarifications. I appreciate digging into the details of [2] and pointing out these subtleties. I agree with your assessment. More details on this in the final version of the paper would be well appreciated. Additional experiments on what happens when learning the noise schedule with fewer steps T and potential "prior hole issues" in connection to that (as discussed in previous comment) would also be nice.
> > > >
> > > > Generally, your replies addressed some initial concerns. Consequently, I raised my score from 6 to 7.
> > > >
> > > > [2] Sohl-Dickstein et al., "Deep Unsupervised Learning using Nonequilibrium Thermodynamics"

---

### Official Review · Reviewer_2LRL · 2021-07-17

**Rating:** 9
**Confidence:** 4

**Summary:**

The paper introduces a family of diffusion-based generative models for density estimations on image datasets. Here, the forward diffusion is directly defined by a set of conditional densities of diffused random variables for a given data. In particular, the paper proposes that $t$-th density for a given data follows a normal distribution; its mean and variance will be determined by data and a monotonic function of $t$, called $SNR(t)$. The generative models aim at learning diffusion processes reversed in time by maximizing its ELBO.

The paper first draws connections between the proposed models and previous diffusion-based models. For example, the authors show that the proposed method's discrete-time versions include previous finite-length Markov chain models as special cases. The paper then demonstrates that the ELBOs of the proposed continuous-time models exist. In here, the resulting ELBOs include an integration (wrt time) of a reconstruction loss weighted by the time-derivative of $SNR(t)$. Based on this, the paper shows that the losses of the previous continuous-time diffusion-based models follow the same integration form but with different weighting functions.

Second, the paper presents interesting properties of the proposed method due to $SNR(t)$. In particular, remind that $SNR(t)$ is invertible because of its monotonicity, and the continuous-time loss's integration contains the time-derivative of $SNR(t)$. The paper shows that the integral can be re-written wrt $SNR(t)$'s output by change-of-variable. In this form, the integral only depends on the range of the integration but doesn't depend on the shape of $SNR(t)$ within the range. As a result, acknowledging the invariant of the loss on the $SNR(t)$'s output space, the paper proposes to parameterize the score models conditioning on log-scaled $SNR(t)$ instead of $t$.

Third, the paper proposes to train $SNR(t)$ for the discrete and continuous-time models. For the monotonicity, $SNR(t)$ is defined as a monotonic network wrt $t$. For discrete-time models, the paper proposes to train $SNR(t)$ networks by maximizing the ELBO together with the diffusion models. For continuous-time models, $SNR(t)$ is trained by minimizing the variance of the Monte-Carlo estimation of the continuous-time ELBOs.

Furthermore, the paper introduces various practical techniques to improve training the models. For example, instead of iid sampling of $t$, the authors propose to use a low-discrepancy sampler of $t$ within a mini-batch. The paper also proposes to use the Fourier feature of inputs: feeding the concatenation of inputs and its Fourier feature to diffusion models.

In the experiments, the paper demonstrates that the proposed models achieve SOTA likelihoods on image datasets, such as CIFAR-10 and ImageNet.

**Limitations And Societal Impact:**

(Limitations)
N/A

(Societal Impact)
N/A

**Main Review:**

(Originality & Significance)
In my understanding, the paper's contributions are clear, and I also consider that the results are essential for several reasons.

First, it introduces a novel family of diffusion-based models whose diffusions are characterized by $SNR(t)$. In particular, with the proposed method, the paper demonstrates the SOTA results on density estimations which autoregressive models have dominated.

Second, the novel $SNR(t)$-based characterization of diffusions helps to understand similar models. For example, the proposed models include some popular diffusion-based models as special cases; thus, one can evaluate the continuous-time ELBOs for those cases. On the other hand, one can apply similar proof techniques for more general cases. Another example includes that for continuous-time cases, the invariance-to-the-noise-schedule property clearly motivates the efficient formulation of score models: models should condition on $SNR(t)$ (or its log-scale), especially when $SNR(t)$ can vary during training.

Third, the paper discusses the benefits of various practical techniques applicable to similar family models with ablation studies. For example, the low-discrepancy sampler of $t$ or Fourier features.

(Quality & Clarity)
In general, the paper has a well-organized structure to motivate the proposed diffusion-based models and other practical techniques to improve training.


(Minor comments)
In Lines 209-210 or Appendix B, I consider some previous works on monotonic neural networks should be cited; for example, J. Sill 1997, C-W. Huang et al. 2018, and A. Wehenkel & G. Louppe 2019.

J. Sill, '97, Monotonic Networks
C-W. Huang et al., '18, Neural Autoregressive Flows
A. Wehenkel & G. Louppe, '19, Unconstrained Monotonic Neural Networks

**Time Spent Reviewing:**

>12hrs

---

> ### Author Response · Authors · 2021-08-10
> **Author Response**
>
> We thank the reviewer for taking their time to read the paper thoroughly, their clear summarization of the main ideas and contributions, and their recognition of the importance and impact of the contributions made.
>
> Regarding your minor comment: we agree that we should cite these earlier works on monotonic networks, and will incorporate these citations in our final draft.

---

### Public Comment · ~Christian_Dietrich_Weilbach1 · 2021-12-02
**Change of variable in eq. (18) for equivalence incorrect?**

I might have overlooked something essential, but by working through the paper I found that $\text{SNR}^{-1}$ does implicitly depend on both $\alpha_t$ and $\sigma_t$ through the forward function $\text{SNR}$ and hence your claim that in equation 18 the integrand does not depend on $\alpha_t$ or $\sigma_t$ is not correct. You hide this dependency by absorbing $\text{SNR}^{-1}$ into $\hat{x}_\theta$ syntactically as $v$ just before the equation. If I missed something important, I would appreciate understanding this better. Thank you!

---

> ### Public Comment · Authors · 2021-12-06
> **the equation is correct, but invariance depends on parameterization of the denoising model**
>
> If I understand your question correctly, you're concerned about our denoising model $\tilde{\mathbf{x}}_{\theta}(\mathbf{z}_v,v)$ implicitly being parameterized in terms of $\alpha_v, \sigma_v$, which would mean that our loss would change as the definition of $\alpha_v, \sigma_v$ changes? This would indeed be true if $\hat{\mathbf{x}}_\theta(\mathbf{z}_t,t)$ is parameterized as a simple function of $t$, since $\tilde{\mathbf{x}}_\theta(\mathbf{z}_v,v) \equiv \hat{\mathbf{x}}_\theta(\mathbf{z}_t,\text{SNR}^{-1}(v))$. In our parameterization of the denoising model we always condition on $(\log(\text{SNR}_\text{max})-\log(\text{SNR}))/(\log(\text{SNR}_\text{max}) - \log(\text{SNR}_\text{min}))$ instead of $t$ itself, so this is not the case in practice. However, we're indeed not explicit about it here and I understand the confusion: It would have been better to be explicit about not parameterizing $\tilde{\mathbf{x}}_\theta(\mathbf{z}_v,v)$ in terms of $\alpha_v, \sigma_v$ in this section.
>
> To be clear to other readers of this comment: Equation 18 is correct for any parameterization of $\tilde{\mathbf{x}}_\theta(\mathbf{z}_v,v)$. Only the dependence of the resulting loss on $\alpha_v, \sigma_v$ depends on the parameterization.
>
> And for those who are curious: We found that the naive parameterization, where $\hat{\mathbf{x}}_\theta(\mathbf{z}_t,t)$ is parameterized as a simple function of $t$, leads to slower optimization, but does not otherwise impact the obtained results.

---

### Public Comment · ~Chin-Yun_Yu1 · 2022-02-05
**Correction of equation 31 and 33**

I found a $\sigma_t$ is missing from equations 31 and 33 when implementing VDM. The correct equations  should be:

$$
\boldsymbol{\mu}_{\boldsymbol{\theta}}(\mathbf{z}_t ; s, t) =\frac{\alpha_s}{\alpha_t}(\mathbf{z}_t+ \sigma_t \operatorname{expm1}(\gamma_\boldsymbol{\eta} (s)-\gamma_\boldsymbol{\eta}(t)) \hat{\boldsymbol{\epsilon}}_\boldsymbol{\theta}(\mathbf{z}_t ; t))
$$
and
$$
\mathbf{z}_s=\sqrt{\alpha_s^2 / \alpha_t^2}(\mathbf{z}_t - \sigma_t c \hat{\boldsymbol{\epsilon}}_\boldsymbol{\theta}(\mathbf{z}_t ; t))+\sqrt{(1-\alpha_s^2) c} \boldsymbol{\epsilon}
$$
respectively.

I actually enjoyed a lot reading this paper. Thank you!

---

> ### Public Comment · ~Diederik_P_Kingma1 · 2022-02-09
> **You are right!**
>
> You are right! Thank you so much for spotting the missing $\sigma_t$'s in Equation 31 and 33 in the Appendix.
>
> Important to note is that (1) these are only used for sampling from the model, and (2) that the experiments were done using the correct formulas, so this does not change the results of the paper.
>
> The correct formulas will be reflected in an upcoming updated arXiv version. Thank you again.

---

### Public Comment · ~David_C_Williams1 · 2022-08-24
**Equation 24**

I would like the authors to check equation 24. I suspect the far RHS is inverted. That is, perhaps the intention was the following:


$$\sigma^{-2}_Q(s,t) = \sigma^{-2}_s + \alpha^2\_{t|s}\sigma^{-2}\_{t|s} = \sigma_t^2/\left(\sigma\_{t|s}^2\sigma_s^2\right)$$

Congratulations on a seminal piece of work.

---

### Decision · Program_Chairs · 2021-09-27

**Decision:**

Accept (Poster)

**Comment:**

This paper presents a novel perspective on denoising diffusion models. The main contributions of this submission include (i) a new formulation of ELBO using $SNR(t)$ and showing invariance to the noise schedule under this formulation, (ii) training noise schedule, and (iii) several new architecture improvements. The paper executes these ideas extremely well and obtains SOTA likelihood results on several images. Although the majority of the improvements in likelihood results seem to be obtained from architectural improvements (i.e., Fourier features), this paper provides very valuable insights into the diffusion models and can be of great interest to the community. Given the contributions, I am pleased to recommend this paper for acceptance.